# RadarQA: Multi-modal Quality Analysis of Weather Radar Forecasts

**Xuming He**[1,2][*][§] **Zhiyuan You**[3][*] **Junchao Gong**[1]**, Couhua Liu**[4]**, Xiaoyu Yue**[1]**,**
**Peiqin Zhuang**[1]**, Wenlong Zhang**[1][†] **Lei Bai**[1][†]

[1] Shanghai Artificial Intelligence Laboratory
[2] ZheJiang University    [3] The Chinese University of Hong Kong
[4] Center for Earth System Modeling and Prediction of China Meteorological Administration
`zhangwenlong@pjlab.org.cn`, `bailei@pjlab.org.cn`

## Abstract

Quality analysis of weather forecasts is an essential topic in meteorology. Although traditional score-based evaluation metrics can quantify certain forecast errors, they are still far from meteorological experts in terms of descriptive capability, interpretability, and understanding of dynamic evolution. With the rapid development of Multi-modal Large Language Models (MLLMs), these models become potential tools to overcome the above challenges. In this work, we introduce an MLLM-based weather forecast analysis method, RadarQA, integrating key physical attributes with detailed assessment reports. We introduce a novel and comprehensive task paradigm for multi-modal quality analysis, encompassing both single frame and sequence, under both rating and assessment scenarios. To support training and benchmarking, we design a hybrid annotation pipeline that combines human expert labeling with automated heuristics. With such an annotation method, we construct RQA-70K, a large-scale dataset with varying difficulty levels for radar forecast quality evaluation. We further design a multi-stage training strategy that iteratively improves model performance at each stage. Extensive experiments show that RadarQA outperforms existing general MLLMs across all evaluation settings, highlighting its potential for advancing quality analysis in weather prediction. The code and dataset are publicly available at https://github.com/hexmSeeU/RadarQA.

## 1   Introduction

Quality analysis of weather forecasts is an essential topic in the field of meteorology [21, 81, 87, 88], playing a critical role in downstream applications such as disaster prevention, risk mitigation, and early warning systems [5, 8, 16]. This analysis evaluates the consistency between predicted and actual weather states, both in single frames and over temporal sequences, aiming to align with the assessment of meteorological experts. Previous methods usually adopt score-based metrics for quality evaluation, which is still far from matching expert-level judgments. First, some descriptive properties (*e.g.*, shape like "scattered and block-like" and movement direction like "moves to the northeast" in Fig. 1) are vital for weather forecasting, but cannot be captured by a simple score. Second, existing methods fail to provide detailed interpretations of the evaluation results, making them less explainable and less convincing. For instance, in Fig. 1, human experts can first observe that "discrepancies arise in shape changes", and then conclude that the forecast's reliability is limited. However, previous score-based metrics lack such interpretive capabilities. Third, human experts can assess the dynamic evolution of weather systems (*e.g.*, "newly formed convective cells are smaller" in Fig. 1), while score-based metrics are primarily limited to pixel-level evaluations of single frames [11, 12, 48], lacking both temporal awareness and global understanding of large-scale weather systems.

---

[*]Equal Contribution.
[†]Corresponding Author.
[§]This work was done during his internship at Shanghai Artificial Intelligence Laboratory.

39th Conference on Neural Information Processing Systems (NeurIPS 2025).

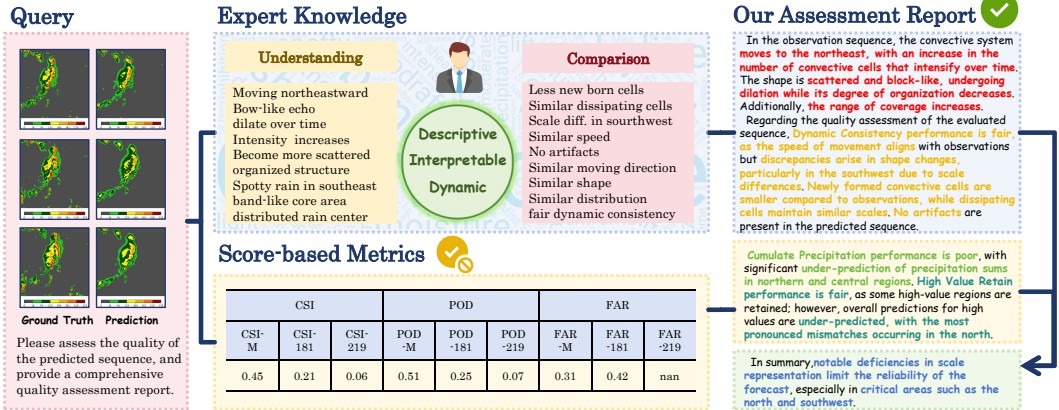

Figure 1: **Comparison of our RadarQA and previous score-based metrics**. Although score-based metrics reveal some forecast deficiencies, such as false alarms, they lack interpretability and sensitivity to dynamics. Our assessment report combines expert knowledge with these metrics, providing a more robust evaluation of the predicted sequence.

To achieve a better weather forecast analysis aligned with human experts, we introduce RadarQA, a multi-modal model for quality analysis of weather radar forecasts. Inspired by the rapid development of MLLMs [3, 10, 32, 40] and MLLM-based image quality assessment methods [34, 73], we believe that descriptive language can effectively incorporate expert knowledge and traditional score-metrics to achieve a more flexible analysis of weather forecast. As shown in Fig. 2d, given a reference sequence and model-generated prediction, RadarQA produces a detailed analysis report from multiple perspectives. First, RadarQA characterizes dynamic properties (*e.g.*, "moves eastward ... block-like structures"). Then, it evaluates the forecast from various angles. For instance, in terms of the *High Value Retain*, the performance is just fair because the high value regions in the north are under-predicted over time, which is a common over-smoothing problem in weather forecast models [17, 18, 61]. Finally, based on the above considerations, RadarQA judges the predicted sequence as poor quality, noting that it "struggles to accurately replicate key features such as scale changes, precipitation distribution, and high-value retention". This evaluation process aligns closely with human experts and offers better interpretability than traditional score-based metrics.

To achieve human-like weather forecast analysis, we propose a set of new and comprehensive tasks. Human experts typically begin by assessing a temporal weather sequence, where single-frame evaluation provides the foundation for sequence assessment. During this process, experts focus on several key factors(*e.g.*, false alarms and misses in a single frame, as well as dynamic consistency and retention of high values in a sequence), integrating them into a detailed assessment report through an interpretation process. To imitate this analysis process, as shown in Fig. 2, we propose a progressive task paradigm consisting of four tasks: (1) Frame Rating, (2) Frame Assessment, (3) Sequence Rating, (4) Sequence Assessment. These tasks meet most common usage scenarios.

To train the expected MLLM, we introduce a comprehensive multi-modal dataset, named RQA-70K. Based on the SEVIR dataset [58], we first implement seven weather nowcasting models to generate model-predicted data. We then carefully design an annotation questionnaire for human experts to annotate 17 key attributes. Besides, we also use scripts to obtain 20 easily computed metric-based attributes. Finally, all these attributes are input into a powerful large language model (*i.e.*, GPT-4o [28]) to generate fluent descriptive languages. To this end, we successfully construct a large-scale, comprehensive dataset, RQA-70K, laying the foundation for model training.

Based on the collected RQA-70K dataset, we further propose a multi-stage training pipeline to train our RadarQA. First, the supervised fine-tuning (SFT) is performed to equip the model with basic task-solving and interpretation capabilities. Second, we design two reward functions and employ reinforcement learning on two rating tasks. This step enhances the model's self-reasoning abilities based on the interpretation abilities acquired from the SFT stage. Third, post fine-tuning is applied with a small subset of samples to further refine performance. Our ablation studies show that this multi-stage training pipeline effectively improves performance on both rating and assessment tasks.

Extensive experiments are conducted to evaluate the effectiveness of RadarQA. First, with the support of RQA-70K, RadarQA outperforms open-source MLLMs by a large margin (*e.g.*, 66.17% *v.s.*

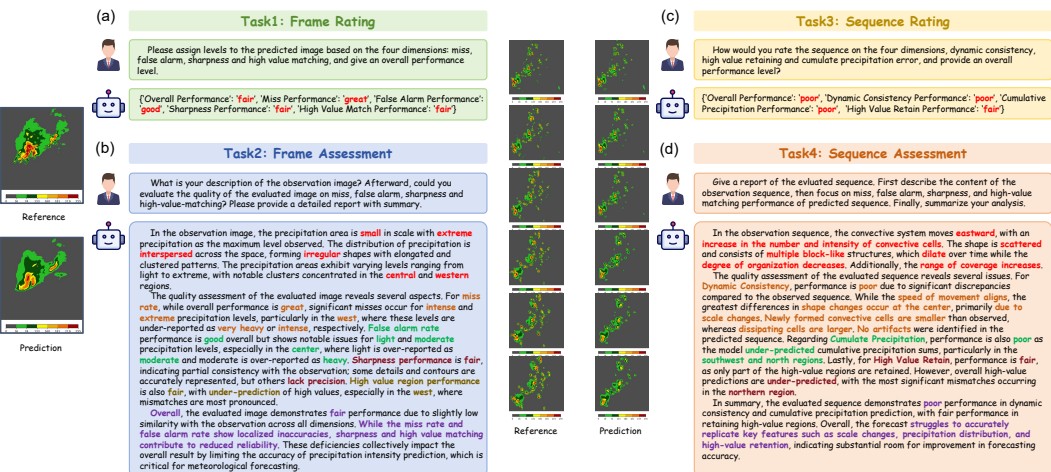

Figure 2: **Task paradigm and qualitative results**. RadarQA focuses on four tasks, including frame rating, frame assessment, sequence rating, and sequence assessment, thereby covering both spatial and temporal modalities, and supporting both quantitative and descriptive evaluations.

36.70% in overall sequence rating). Second, our RadarQA can generate a detailed and comprehensive assessment report, as shown in Fig. 2, even surpassing the powerful OpenAI o1 [29] (6.58 *v.s.* 5.49 in GPT-4 Score for sequence assessment). These results demonstrate the superiority of RadarQA and highlight the research potential of multi-modal weather forecast analysis tasks. Finally, experiments on the out-of-distribution radar data synthesis task further verify the effectiveness of RadarQA.

## 2    Related Works

**Quality assessment of weather forecast** leverages verification metrics to evaluate the accuracy and reliability of weather predictions [13, 14, 31, 43–45, 54, 63, 64]. For example, the Critical Success Index (CSI) [13], a traditional categorical metric, measures the ratio of correctly predicted events to the total forecasted and observed events, penalizing both false alarms and missed detections. In contrast, the Structural Similarity Index Measure (SSIM) [63], originally developed for general image quality assessment, has been adapted to evaluate the consistency of spatial patterns in weather forecasts. However, as stated in Sec. 1, these score-based metrics do not fully align with human experts, particularly in terms of descriptive properties, interpretation process, and the perception of dynamic evolution, making them far from being satisfactory in real-world applications.

**Multi-modal Large Language Models** (MLLMs) extend Large Language Models (LLMs) [6, 20, 57, 70] by integrating other modalities, particularly vision, to enable unified understanding across different input types. Recent advances in MLLMs [3, 10, 32, 33, 35, 40, 60, 68, 69, 71, 72, 79] have led to superior performance on a wide range of tasks, including image captioning [1, 9, 37, 56, 76], visual question answering [2, 23, 39, 42, 51, 59, 89, 86], and multi-step reasoning [52, 62]. However, the weather forecast analysis ability of these MLLMs is still limited, as shown in Sec. 5.

**MLLM-based quality assessment** utilizes the power of MLLMs to conduct visual quality assessment across diverse modalities, including images [15, 34, 65–67, 73–75, 85], videos [19, 30, 83] and 3D point clouds [84]. For instance, Q-Insight [34] employs Group Relative Policy Optimization (GRPO) [53] to guide models in reasoning across different tasks. Q-Bench-Video [83] incorporates a diverse set of videos to assess the video quality through various Question-Answer (QA) formats. LLM-PCQA [84] designs a novel prompt structure that enables MLLMs to perceive the point cloud visual quality. However, the potential of MLLMs in weather forecast quality analysis is still under-explored.

## 3    Task Paradigm and Dataset Construction

### 3.1    Task Paradigm

Meteorological experts typically construct a comprehensive reasoning chain based on both quantitative metrics and expert visual perception of convective structures to evaluate weather forecasting results.

By examining discrepancies between the ground truth and predictions, experts incorporate prior knowledge, such as domain expertise, to provide a quality analysis of the predictions. To align with this expert evaluation process, as highlighted in Sec. 1, we aim to establish a multi-functional, multi-modal, and multi-dimensional task paradigm for quality analysis of weather radar forecast scenarios. Specifically, our RadarQA should possess the following abilities:

*Ability-1*. RadarQA is required to evaluate differences in dynamic properties across the entire sequence over time, as in Fig. 2c, d. Considering that single-frame analysis is the basis of sequence analysis, RadarQA also needs to analyze the quality of individual frames (*e.g.*, tasks in Fig. 2a, b).

*Ability-2*. RadarQA is required to rate different general attributes, and to integrate these ratings into an overall quality rating. This reflects that meteorological experts assist their evaluations by considering a combination of diverse general attributes (*e.g.*, rating tasks in Fig. 2a, c).

*Ability-3*. RadarQA should be capable of generating high-quality evaluation reports for predictions. This mirrors the real-world workflow where meteorologists compose comprehensive reports for the forecasting department after forming a brief judgment (*e.g.*, assessment tasks in Fig. 2b, d).

To reflect the above abilities, we establish a task paradigm with the following four tasks to progressively guide MLLMs toward expert-like analysis:

*Task-1: Frame Rating*. As shown in Fig. 2a, given a model-predicted image and its corresponding ground truth image, the model should assign discrete rating levels for four static general attributes: *Miss*, *False Alarm*, *Sharpness*, and *High Value Match*, each reflecting a specific aspect of the prediction quality. These are then followed by an *Overall* performance that summarizes the general quality.

*Task-2: Frame Assessment*. In addition to provide discrete ratings, the model should generate qualitative descriptions outlining both correctly predicted features and notable deficiencies with respect to some key attributes (*e.g.*, "significant misses occur for intense and extreme precipitation levels" in Fig. 2b, detailed below), and explain how these attributes affect the overall prediction.

*Task-3: Sequence Rating*. As illustrated in Fig. 2c, given a forecasted sequence, the model is expected to assign quality ratings for three dynamic general attributes: *Dynamic Consistency*, *Cumulative Precipitation*, *High Value Retain*, followed by an *Overall* quality rating.

*Task-4: Sequence Assessment*. Building upon the sequence rating levels and additional key sequence attributes (detailed below), the model should first provide a comprehensive description of the performance for each dynamic general attributes (*e.g.* "Newly formed convective cells are smaller than observed" in Fig. 2d), then summarize how these dimensions affect the overall performance.

## 3.2 Scientific Attribute Library

As stated in Sec. 3.1, several key attributes are needed in our task paradigm. Existing evaluation attributes, such as Critical Success Index (CSI) [13] and Probability of Detection (POD) [45], assess prediction quality from various perspectives at the pixel level. Although these metrics capture certain characteristics of weather forecast scenarios, they fall short in identifying discrepancies at the structural level, especially from the perspective of physically grounded convective weather systems. Moreover, existing approaches often overlook the temporal dynamics inherent in forecast sequences, which are crucial for analyzing the evolution of physical patterns. To address these limitations, we aim to develop a comprehensive scientific attribute library that integrates physics-informed attributes into the quality analysis framework.

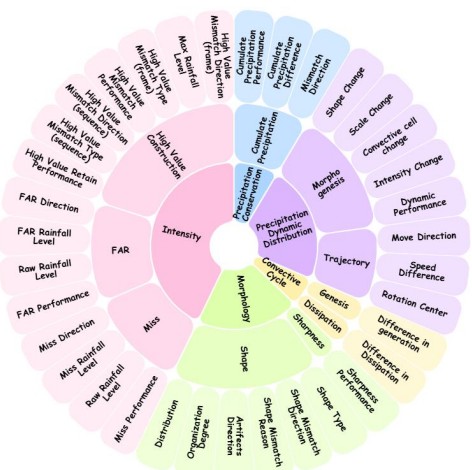

Figure 3: **Overview of our scientific attribute library** with 5 super-categories and 10 sub-categories in total.

**Attribute library**. As illustrated in Fig. A5, our attribute library is organized into five super-categories, encompassing fundamental physical attributes such as *morphology* and *intensity*, atmospheric physics properties like *rainfall conservation* and *convective cycle*, as well as temporal characteristics *precipitation dynamic distribution*. Each super-category comprises multiple sub-categories, from which we identify key attributes that cover both frame-level

and sequence-level features. These attributes are then used to guide the dataset construction. In total, we define *15 frame attributes* and *22 sequence attributes*. See details in the Appendix.

**General attributes used in rating tasks**. Under the guidance of domain experts, we identify seven general evaluation attributes. These general attributes are used in rating tasks, while all attributes are used in assessment tasks as stated in Sec. 3.3. The general attributes are derived by refining existing score-based metrics and integrating perception-based attributes. The definitions of these attributes are detailed below. (a) *Miss*. The proportion of convective regions in the ground truth that are not captured by the prediction. (b) *False Alarm*. The proportion of predicted convective regions that do not correspond to any actual event in the ground truth. (c) *Sharpness*. The degree to which the predicted convective structures maintain clear, well-defined boundaries. (d) *High Value Match*. The extent to which the core regions of convective systems, *i.e.*, high-intensity areas, in the prediction align with the high-intensity regions in the ground truth. (e) *Dynamic Consistency*. The ability of the model to accurately capture the evolution of convective systems over time, including factors such as the movement speed, the genesis of convection, and the dissipation of convective cells. (f) *Cumulative Precipitation*. The ability of the model to reproduce the temporally integrated precipitation amounts associated with convective systems. (g) *High Value Retain*. The ability of the model to preserve high-intensity regions throughout temporal evolution.

### 3.3 Dataset Construction

High-quality and large-scale datasets are crucial for training MLLMs to conduct reliable quality analysis. Although post-training techniques such as GRPO [53] have shown promising capabilities in enhancing model performance with limited data, it remains essential to first empower the model with intensive and diverse data to ensure baseline competency for the target task. In this section, we elaborate on the construction of our dataset, covering forecast data collection, query collection, and response generation. An overview of the dataset construction pipeline is shown in Fig. 4.

**Forecast data collection**. As shown in Fig. 4, we construct the RawRQA-20K dataset based on the widely used SEVIR dataset [58], which encompasses a wide range of events, including various types of storm events and random phenomena. For our task, we focus exclusively on storm events to build the dataset, covering thunderstorm wind, flood, flash flood, funnel cloud, hail, heavy rain, and tornado. The strong convective nature of these events poses greater forecasting challenges and thus provides higher value for analysis. We focus on the Vertically Integrated Liquid (VIL) modality and split each storm event into three input-target pairs, where each input consists of 10 consecutive frames and each target consists of the following 12 frames, thus forming a specialized SEVIR subset.

For sequence prediction, we adopt a variety of weather prediction models to generate diverse predicted sequences. These models include EarthFormer [17], PredRNN [61], Cascast [22], DGMR [47], Diffcast [77], Simvp [18], and Nowcastnet [82], covering a wide range of model architectures, including generative adversarial networks, recurrent neural networks, and diffusion models.

With these model-predicted sequences, we apply VIL discretization and colorization to render the radar data into RGB space. Following [22, 49, 77, 82], we categorize the VIL values into six precipitation levels reflecting different intensities of convective activity. We then apply the colormap provided by SEVIR to the generated prediction sequences for visualization, resulting in our raw prediction dataset, RawRQA-20K. Additionally, to conduct quality analysis on single frames, we randomly select one frame from each prediction sequence in RawRQA-20K and pair it with the corresponding ground truth frame. Together, these two data modalities enable a comprehensive evaluation of both static and dynamic properties within individual frames and sequences, respectively.

**Query collection**. Following [73, 74], we leverage GPT-4o [28] to generate 50 candidate questions for both the brief and detailed tasks. Based on syntactic structure, lexical diversity, and overall clarity, we manually select a set of 10 questions that are both clear and varied. During training and evaluation, these questions are randomly sampled to construct data tuples for model input.

**Response collection**. As shown in Fig. 2, we employ two types of responses. The first comprises concise, structured outputs for rating tasks, while the second consists of detailed quality reports for assessment tasks. For detailed responses, existing methods primarily rely on either human annotation [67] or generation by MLLMs [66, 74]. However, human annotations often vary in quality [74], and MLLMs remain unreliable for meteorological tasks [7, 41], as evidenced by the results in Sec. 5.

We propose an **Attribute-Informed Generation** method to enable effective annotation for detailed responses. We observe that key attributes can often be decoupled within evaluation responses

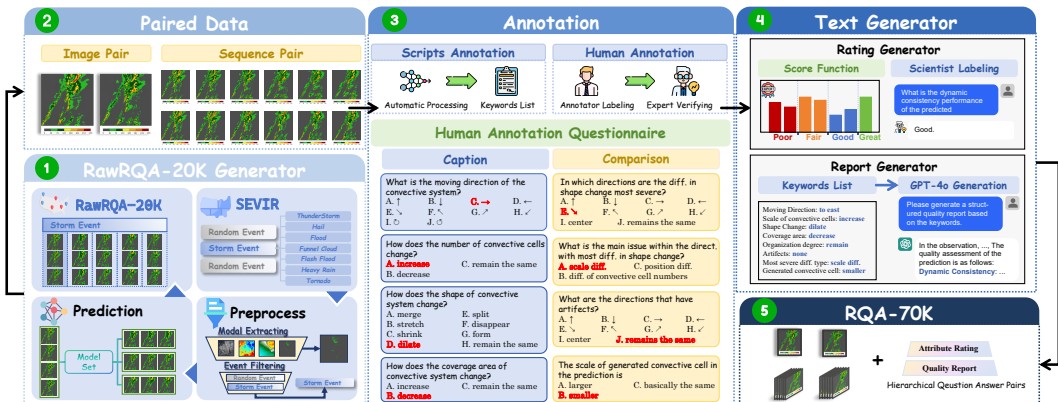

Figure 4: **Construction of our RQA-70K dataset**. First, *RawRQA-20K Generator* produces frame and sequence samples based on the SEVIR dataset. Next, we annotate the data using script functions and the *Human Annotation Questionnaire*. Then, *Text Generator* produces corresponding responses from the annotated attributes, which are paired with question templates to construct RQA-70K.

constructed by human experts. Inspired by this insight, given a set of annotated key attributes, we leverage them to produce highly informative quality assessment reports, as shown in the *Text Generator Module* part of Fig. 4. For rating tasks, we automate the generation of JSON-formatted responses based on the general attributes outlined in Sec. 3.1. For assessment tasks, all frame or sequence attributes from the key attribute database are provided to GPT-4o to generate detailed assessment reports. To ensure the reliability of the generated response, we also provide GPT-4o with all relevant visual information and explicitly instruct it to correct potential inconsistencies.

Under the *Attribute-Informed Generation* framework, the focus of dataset construction shifts to attribute annotation. All attributes are categorized into two types: 17 perception-based and 20 metric-based attributes, whose annotation processes are detailed below.

*Perception-based attributes* involve the understanding of visual content and convective structures, which requires expert knowledge for reliable annotation. Therefore, we employ human annotation to ensure high-quality labeling, as shown in the *Human Annotation Questionnaire* in Fig. 4. The questionnaire consists of two types of questions: one focuses on understanding the observation (*i.e.*, the *caption* part), and the other evaluates the quality of predictions (*i.e.*, the *comparison* part). First, experts define labeling guidelines, construct golden standards, and provide reference samples. Second, using these samples, annotators are guided to align with domain experts through pilot testing and iterative refinement to ensure annotation quality. Third, once annotators meet alignment criteria, they proceed to large-scale labeling, during which experts conduct random checks to ensure consistency. If a batch passes validation, it is included in the key attribute database; otherwise, it is returned for re-annotation until the quality standards are met. More details are provided in the Appendix.

*Metric-based attributes* require precise numerical values. We use the script function to annotate and involve experts in setting key parameters. See Appendix for details.

**Dataset statistics**. The statistics of our dataset are summarized in Tab. 1. Our dataset consists of 40,000 brief templated samples (training set of rating tasks), along with 29,000 detailed, high-quality samples (training set of assessment tasks). To ensure the reliability of these samples, all annotations undergo expert validation, and automated annotations are routinely verified through expert spot-checking on sampled batches to ensure accuracy.

Table 1: **Statistics** of our RQA-70K dataset.

|  | Task-1 *Frame Rating* | Task-2 *Frame Assessment* | Task-3 *Sequence Rating* | Task-4 *Sequence Assessment* |
|---|---|---|---|---|
| Train | 20,000 | 14,500 | 20000 | 14500 |
| Validation | 860 | 410 | 801 | 179 |

## 4 Model Training

Inspired by [24], we adopt a multi-stage training strategy to progressively adapt the model to the domain-specific tasks. In Stage 1, we perform supervised fine-tuning on large-scale multimodal data to equip the model with basic task-solving capabilities. In Stage 2, we use reinforcement learning [50, 53, 78] and carefully design two reward functions for the rating tasks. We encourage the model to reason based on the interpretation abilities acquired from Stage 1. In Stage 3, we apply

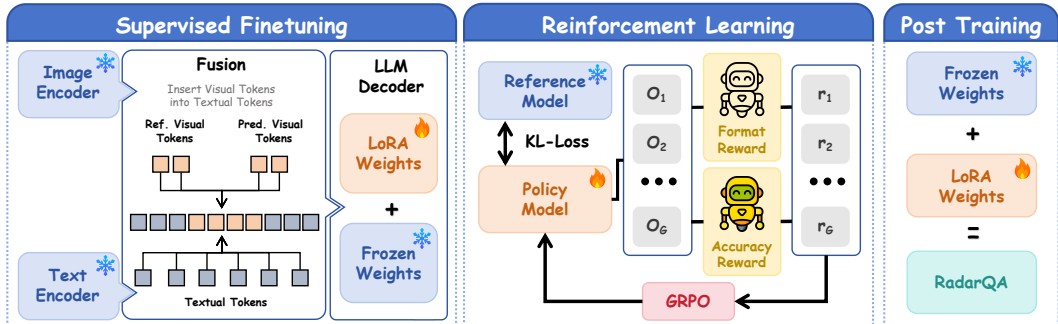

Figure 5: **Training pipeline of our RadarQA**. First, we apply supervised fine-tuning with LoRA on RQA-70K to equip the model with basic capabilities. Then, GRPO is used to enhance performance on rating tasks by leveraging its learned assessment ability. Finally, post-training is applied to standardize output formats and further improve overall performance.

post-training with a small set of samples to further refine performance. An overview of our training pipeline is shown in Fig. 5. We validate the effectiveness of our multi-stage training strategy in Sec. 5.3, which demonstrates consistent performance improvements at each stage.

**Stage 1: Supervised fine-tuning**. We employ RQA-70K for supervised fine-tuning in this stage. Since full LLM fine-tuning is highly computationally demanding and requires large-scale datasets, we adopt LoRA [27], a parameter-efficient fine-tuning method that injects trainable low-rank matrices into certain layers while keeping most original parameters frozen, to address the issue of limited data.

**Stage 2: Reinforcement learning**. Inspired by [34], we adopt GRPO [53] in the second stage to optimize the model's performance on the *rating tasks*. In this phase, the fine-tuned model from Stage 1 serves as the policy model to be further refined. Since GRPO requires well-defined reward functions to guide policy updates, we introduce two task-specific rewards. (a) *Format Reward*. The model is required to generate responses in a well-structured JSON format, where each key corresponds to a general attribute of the brief task. Denoting the format reward as $r_{fmat}$. If the response can be successfully parsed into a valid JSON object and all required keys are present, we set $r_{fmat} = 1$. Otherwise, the reward is 0. (b) *Accuracy Reward*. If the response generated by the policy model can be correctly parsed into a valid JSON format, we compare the predicted performance levels for each general attribute with the corresponding ground truth labels. Let $N_{all}$ be the total number of general attributes and $N_{hit}$ the number of correctly predicted general attributes. The accuracy reward is defined as $r_{acc} := N_{hit}/N_{all}$ if $r_{fmat} = 1$; otherwise, it is set to 0.

**Stage 3: Post-training**. To further refine model performance, we conduct post-training in this stage by using a small subset of RQA-70K, applying low-rank LoRA updates for effective adaptation.

## 5 Experiments

### 5.1 Details and Metrics

**Implementation details**. We adopt Qwen-2.5-VL-7B [3] as the base model. In Stage 1, we employ AdamW as the optimizer, with an initial learning rate of $1 \times 10^{-4}$. We integrate LoRA with a rank of 8, The model is trained with a total batch size of 128 for 5 epochs on RQA-70K. In Stage 2, we set the generation number of GRPO to 4, and train the model for 1 epoch on 10,000 randomly selected brief task samples with a total batch size of 32. In Stage 3, we set the LoRA rank to 4 and fine-tune the model for 1 epoch using 2,500 samples from each sub-task. The entire training process takes approximately 50 hours using 8 NVIDIA A800 GPUs.

**Metrics**. For the rating tasks, we adopt accuracy as the evaluation metric. Specifically, we prompt MLLMs to generate responses in a structured JSON format with predefined keys. Accuracy is then computed separately for each general attributes. For the assessment tasks, we employ standard metrics, including BERTScore [80], BLEU [46], ROUGE_L [36], and METEOR [4]. Following [38, 73], we also incorporate the GPT-4 score, where the model's response is rated from 0 to 10 based on relevance, accuracy, and level of detail with respect to the ground truth.

Table 2: **Results** on general attributes for the frame rating and frame assessment tasks. Accuracy is used as the metric for the frame rating task. RadarQA surpasses all baselines by a large margin.

| Methods | | Frame Rating | | | | | Frame Assessment | | | | |
|---|---|---|---|---|---|---|---|---|---|---|---|
| | | Overall | False Alarm | Miss | High Value | Sharpness | BLEU | BERTScore | ROUGE_L | METEOR | GPT4Score |
| Open Source | Qwen2.5-VL-7B | 20.10 | 36.40 | 30.00 | 16.51 | 35.93 | 0.122 | 0.750 | 0.389 | 0.332 | 3.81 |
| | InternVL2.5-8B | 30.89 | 21.86 | 8.95 | 1.04 | 36.51 | 0.114 | 0.745 | 0.426 | 0.335 | 3.50 |
| | Qwen2.5-VL-72B | 23.76 | 27.72 | 40.59 | 6.93 | 39.60 | 0.132 | 0.749 | 0.396 | 0.324 | 4.32 |
| API-based | GPT4o | 48.84 | 31.40 | 23.85 | 11.04 | 52.91 | 0.116 | 0.760 | 0.408 | 0.345 | 5.27 |
| | Claude3.7 sonnet | 39.77 | 32.79 | 27.21 | 21.74 | 43.14 | 0.083 | 0.754 | 0.377 | 0.350 | 5.89 |
| | Gemini2.5 pro | 21.40 | 29.65 | 31.16 | 29.30 | 40.58 | 0.080 | 0.741 | 0.348 | 0.326 | 5.77 |
| | o1 | 52.67 | 28.86 | 23.83 | 28.15 | 50.58 | 0.091 | 0.739 | 0.330 | 0.288 | 5.63 |
| Ours | RadarQA | **61.51** | **65.35** | **67.67** | **69.19** | **78.60** | **0.213** | **0.809** | **0.512** | **0.420** | **6.87** |

Table 3: **Results** on general attributes for the sequence rating and sequence assessment tasks. Accuracy is used as the metric for the sequence rating task. RadarQA achieves the best performance.

| Methods | | Sequence Rating | | | | | Sequence Assessment | | | | |
|---|---|---|---|---|---|---|---|---|---|---|---|
| | | Overall | Dynamic Consistency | Cumulate Precipitation | High Value Retain | BLEU | BERTScore | ROUGE_L | METEOR | GPT4Score |
| Open Source | Qwen2.5-VL-7B | 7.99 | 16.10 | 17.49 | 23.22 | 0.090 | 0.745 | 0.281 | 0.342 | 3.92 |
| | InternVL2.5-8B | 36.70 | 40.20 | 31.46 | 21.10 | 0.010 | 0.636 | 0.241 | 0.251 | 2.61 |
| | Qwen2.5-VL-72B | 19.80 | 46.53 | 23.76 | 7.92 | 0.132 | 0.740 | 0.329 | 0.335 | 4.72 |
| API-based | GPT4o | 45.00 | 22.60 | 26.59 | 4.99 | 0.11 | 0.757 | 0.323 | 0.369 | 4.39 |
| | Claude3.7 sonnet | 19.48 | 26.22 | 21.10 | 14.48 | 0.052 | 0.737 | 0.266 | 0.337 | 5.56 |
| | Gemini2.5 pro | 27.59 | 28.34 | 26.72 | 22.47 | 0.055 | 0.739 | 0.254 | 0.341 | 5.63 |
| | o1 | 29.70 | 33.66 | 29.70 | 19.80 | 0.091 | 0.733 | 0.254 | 0.304 | 5.49 |
| Ours | RadarQA | **66.17** | **53.31** | **48.94** | **80.52** | **0.212** | **0.815** | **0.436** | **0.461** | **6.58** |

## 5.2 Experimental Results

**Quantitative results of frame rating task** are shown in Tab. 2. First, the performance of open-source MLLMs remains limited. In particular, for the *High Value Match* attribute, all three open-source baselines achieve accuracies below 20%, indicating that they still struggle to associate different rainfall intensities with the corresponding color mappings. Second, among the API-based methods, o1 outperforms other models under the same evaluation setting. Finally, RadarQA significantly surpasses all baseline methods, demonstrating the superior effectiveness of our approach.

**Quantitative results of frame assessment task** are illustrated in Tab. 2. First, open-source models exhibit clear limitations on the more challenging frame assessment task; their relatively low GPT-4 scores indicate a lack of domain-specific understanding. Second, among the API-based models, Gemini 2.5 Pro achieves the best overall performance. Finally, RadarQA outperforms all baselines across all metrics, demonstrating its superior ability to capture and interpret convective features.

**Quantitative results of sequence rating task** are demonstrated in Tab. 3. First, among open-source models, Intern-VL-2.5-8B [10] achieves the best performance, even surpassing the larger Qwen-VL-2.5-72B [3]. Second, API-based models consistently exhibit limited capability on sequence rating, with average accuracies ranging between 20% and 30%. Finally, RadarQA outperforms all baseline methods, particularly achieving over 80% accuracy on the *High Value Retain* attribute.

**Quantitative results of sequence assessment task** are shown in Tab. 3. First, compared to frame assessment, both open-source and API-based models perform worse on sequence assessment, indicating that understanding and assessing sequences is more challenging. This is primarily due to two factors. (a) The inherent complexity of video modality, which requires analyzing temporal correlations across frames. (b) The construction of ground truth responses based on a large number of expert-annotated attributes, which involve various meteorological concepts such as "Convection Genesis" in Fig. A5. Second, RadarQA still achieves excellent performance, highlighting its superior capabilities in interpreting temporal information.

**Qualitative results of assessment tasks** are illustrated in Fig. 2 and Fig. 1. First, RadarQA effectively captures the dynamic evolution of convective systems (*e.g.*, "dilating over time while the degree of organization decreases" in Fig. 2). Second, RadarQA can also interpret key deficiencies across multiple dimensions (*e.g.*, "struggles to accurately replicate key features such as scale changes" in Fig. 2). Additional qualitative results for assessment tasks are provided in the Appendix.

Table 4: **Ablation studies of our multi-stage training strategy**. Frame / sequence rating tasks are evaluated in average accuracy, while frame / sequence assessment tasks are assessed in GPT-4 Score. Our full 3-stage training pipeline achieves the best results.

| # | Stage-1 | Stage-2 | Stage-3 | Rating | Assessment |
|---|---------|---------|---------|--------|------------|
| 0 | ✗ | ✗ | ✗ | 27.79 / 16.20 | 3.81 / 3.92 |
| 1 | ✓ | ✗ | ✗ | 64.05 / 55.15 | 6.40 / 6.22 |
| 2 | ✓ | ✓ | ✗ | 66.95 / 61.58 | -[a] |
| 3 | ✓ | ✗ | ✓ | 68.14 / 62.17 | 6.83 / 6.56 |
| 4 | ✓ | ✓ | ✓ | **68.46 / 62.24** | **6.87 / 6.58** |

[a]Stage-2 is trained only on rating tasks.

Table 6: **Ablation studies of multi-dataset joint training**. Training on four tasks outperforms training on each task. Metrics are average accuracy (Task-1 & Task-3) and GPT-4 Score (Task-2 & Task-4). For single-task training, each task is trained on its corresponding dataset.

| Training data | Task-1 | Task-2 | Task-3 | Task-4 |
|---------------|--------|--------|--------|--------|
| Single-task data | 66.63 | 6.48 | 59.55 | 6.17 |
| All-task data | **68.46** | **6.87** | **62.24** | **6.58** |

Table 5: **Results on out-of-distribution task**. RadarQA is requested to evaluate radar reflectivity reconstruction task, which is *unseen* during training. Frame / sequence rating tasks are evaluated in average accuracy, while frame / sequence assessment tasks are assessed in GPT-4 Score.

| Methods | | Rating | Assessment |
|---------|---|--------|------------|
| Open Source | Qwen-2.5-VL-72B | 27.72 / 21.15 | 3.60 / 3.78 |
| API-based | GPT-4o | 23.17 / 23.71 | 4.30 / 3.82 |
| | o1 | 32.28 / 17.31 | 4.34 / 4.36 |
| Ours | RadarQA | **59.94 / 48.72** | **6.22 / 5.64** |

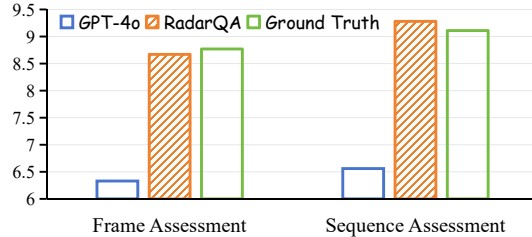

Figure 6: **Expert Study** of frame assessment and sequence assessment tasks.

**Results on out-of-distribution task** are illustrated in Tab. 5. We employ three models designed for radar reflectivity reconstruction, including DiffSR [25], SRViT [55], and U-Net [26], to generate out-of-distribution (OOD) samples on a different dataset for evaluation. For both the frame rating and assessment tasks, RadarQA maintains high accuracy even under the challenging OOD setting and significantly outperforms the baseline methods. For sequence rating and assessment tasks, although performance declines to some extent, RadarQA still surpasses all baselines by a notable margin. This performance gap is primarily due to the lack of explicit temporal modeling in radar reflectivity reconstruction. When each frame in a sequence is predicted independently, the resulting sequence lacks temporal coherence, which may hinder the model's ability to make consistent assessments.

**Expert study**. To evaluate the alignment between RadarQA and human experts, we invited meteorologists to rate the ground truth, RadarQA, and GPT-4o on the assessment tasks on three criteria: content accuracy, information density, and coverage of expert-concerned issues. As shown in Fig. 6, both the ground truth and RadarQA outperform GPT-4o, confirming the effectiveness of the task design and the strong performance of RadarQA. Moreover, scores on the sequence assessment task are generally higher than those on the frame assessment task, highlighting the value of integrating expert knowledge into the assessment process.

## 5.3 Ablation Studies

**Training strategy**. To enhance model performance, we adopt a multi-stage training pipeline (see Fig. 5) comprising supervised fine-tuning, reinforcement learning, and post-training. To evaluate the effectiveness of each stage, we compare models trained with different combinations of the three training stages. First, after the Stage 1 training, the model demonstrates a relative improvement of 40% in average accuracy on rating tasks and achieves around 2.5-point increase in GPT-4 Score on assessment tasks, indicating enhanced domain understanding (*i.e.*, #0 *v.s.* #1 in Tab. 4). Second, combining Stage 1 with either Stage 2 or Stage 3 yields further improvements over using Stage 1 alone (*i.e.*, #2 & #3 in Tab. 4). Finally, as shown in #4 in Tab. 4, the full training pipeline achieves the best performance across all four tasks, demonstrating the effectiveness of our training strategy.

**Joint training on multiple tasks**. To demonstrate the effectiveness of multi-task training, we compare our jointly trained RadarQA with four single-task variants, each trained separately on a specific task. As shown in Tab. 6, RadarQA consistently outperforms all single-task models across their respective metrics, highlighting the overall efficacy of our multi-task training approach.

# 6 Conclusions and Limitations

We introduce RadarQA, an MLLM-based model for quality analysis of weather radar forecasts. Empowered by a novel task paradigm, a high-quality dataset RQA-70K, and a multi-stage training pipeline, RadarQA outperforms all baseline methods across all tasks and under out-of-distribution settings, demonstrating potential for advanced applications in meteorology.

**Limitations**. First, our task paradigm is not yet fully unified. Extending the framework to support comparisons between two predicted results can further enhance practicality. Second, the fine-grained descriptions are still not satisfactory. Finally, whether the assessment outputs can serve as feedback or rewards to improve forecasting models remains underexplored. These are left for future work.

# 7 Acknowledgements

This work is Supported by Shanghai Artificial Intelligence Laboratory.

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

# Appendix

## A  Overview

This Appendix is structured as follows. Dataset details are described in Appendix B. More ablation studies, qualitative and quantitative results are presented in Appendix C

## B  Dataset Details

### B.1  Details of Scientific Attribute Library

To facilitate dataset construction, we design a scientific attribute library grounded in physical principles. This library comprises 5 super-categories and 10 sub-categories, comprising 35 attributes. Combined with the overall performance of the predictions at both the frame and sequence levels, these constitute a total of 37 key attributes used for dataset construction. The definitions of the 35 attributes in our scientific attribute library are provided in detail below.

**Intensity**.

- **Miss**. (a) Miss Performance. The proportion of regions with observed precipitation in the ground truth that are incorrectly predicted as "sunny" in the forecast. (b) Raw Rainfall Level. The rainfall levels in the ground truth for regions where rainfall is missed in the prediction. (c) Miss Rainfall Level. The rainfall levels in the prediction for regions where rainfall is missed. (d) Miss Direction. The directions in the prediction in which specific rainfall levels that are missed in the prediction.

- **FAR**. (a) FAR Performance. The proportion of regions labeled as "sunny" in the ground truth but incorrectly predicted with precipitation. (b) Raw Rainfall Level. The rainfall levels in the ground truth for regions where rainfall is falsely alarmed. (c) FAR Rainfall Level. The rainfall levels in the prediction for regions where rainfall is falsely alarmed. (d) FAR Direction. The directions in the prediction in which specific false-alarm rainfall levels that appear in the prediction.

- **High Value Construction**. (a) High Value Retain Performance. The ability of the prediction to consistently preserve high-value regions. (b) High Value Mismatch Type (Sequence). The type of mismatch in regions with high values (*i.e.*, precipitation at "intense" level or above) across the prediction and ground truth sequence. (c) High Value Mismatch Direction (Sequence). The directions in which high-value regions were mismatched. (d) High Value Mismatch Performance. The ability of the prediction to predict intense precipitation levels. (e) High Value Mismatch Type (Frame). The type of mismatch in regions with high values (*i.e.*, precipitation at "intense" level or above) across the prediction and ground truth frame. (f) High Value Mismatch Direction (Frame). The directions in which high-value regions were mismatched. (g) Max Rainfall Level. The maximum precipitation level in the observation.

**Precipitation Conservation**.

- **Cumulate Precipitation**. (a) Cumulate Precipitation Performance. The degree to which the cumulative precipitation predicted over the entire sequence aligns with the ground truth. (b) Cumulate Precipitation Difference. Differences between the total precipitation of the prediction and the ground truth across the sequence, indicating whether the forecast overestimates or underestimates cumulative rainfall. (c) Mismatch Direction. The directions in which the prediction fails to reconstruct the cumulative precipitation accurately.

**Precipitation Dynamic Distribution**.

- **Morphogenesis**. (a) Shape Change. The change in the shape of the convective system over time in the ground truth. (b) Scale Change. The change in the spatial area of the convective system across frames in the ground truth. (c) Convective Cell Change. The change in the number of convective cells. (d) Intensity Change. The change in the precipitation intensity over time. (e) Dynamic Consistency Performance. The overall consistency of dynamic evolution between the prediction and the ground truth.

- **Trajectory**. (a) Move Direction. The primary direction of movement of the convective system in the ground truth. (b) Speed Difference. The difference in the movement speed of the convective

Table A1: **Characteristics** of each attribute in terms of level (frame / sequence), reference type (caption / comparison), annotation method (human / automation), and usage purpose (rating / assessment).

| Attributes | Level | | Reference | | Annotation | | Usage | |
|---|---|---|---|---|---|---|---|---|
| | Frame | Sequence | Caption | Comparison | Human | Automation | Rating | Assessment |
| Miss Performance | ✓ | ✗ | ✗ | ✓ | ✗ | ✓ | ✓ | ✓ |
| Raw Rainfall Level for Miss | ✓ | ✗ | ✗ | ✓ | ✗ | ✓ | ✗ | ✓ |
| Miss Rainfall Level | ✓ | ✗ | ✗ | ✓ | ✗ | ✓ | ✗ | ✓ |
| Miss Direction | ✓ | ✗ | ✗ | ✓ | ✗ | ✓ | ✗ | ✓ |
| FAR Performance | ✓ | ✗ | ✗ | ✓ | ✗ | ✓ | ✓ | ✓ |
| Raw Rainfall Level for FAR | ✓ | ✗ | ✗ | ✓ | ✗ | ✓ | ✗ | ✓ |
| FAR Rainfall Level | ✓ | ✗ | ✗ | ✓ | ✗ | ✓ | ✗ | ✓ |
| FAR Direction | ✓ | ✗ | ✗ | ✓ | ✗ | ✓ | ✗ | ✓ |
| High Value Retain Performance | ✗ | ✓ | ✗ | ✓ | ✗ | ✓ | ✓ | ✓ |
| High Value Mismatch Type (sequence) | ✗ | ✓ | ✗ | ✓ | ✗ | ✓ | ✗ | ✓ |
| High Value Mismatch Direction (sequence) | ✗ | ✓ | ✗ | ✓ | ✗ | ✓ | ✗ | ✓ |
| High Value Mismatch Performance | ✓ | ✗ | ✗ | ✓ | ✗ | ✓ | ✓ | ✓ |
| High Value Mismatch Type (Frame) | ✓ | ✗ | ✗ | ✓ | ✗ | ✓ | ✗ | ✓ |
| High Value Mismatch Direction (Frame) | ✓ | ✗ | ✗ | ✓ | ✗ | ✓ | ✗ | ✓ |
| Max Rainfall Level | ✓ | ✗ | ✓ | ✗ | ✗ | ✓ | ✗ | ✓ |
| Cumulate Precipitation Performance | ✗ | ✓ | ✗ | ✓ | ✗ | ✓ | ✓ | ✓ |
| Cumulate Precipitation Difference | ✗ | ✓ | ✗ | ✓ | ✗ | ✓ | ✗ | ✓ |
| Mismatch Direction | ✗ | ✓ | ✗ | ✓ | ✗ | ✓ | ✗ | ✓ |
| Shape Change | ✗ | ✓ | ✓ | ✗ | ✓ | ✗ | ✗ | ✓ |
| Scale Change | ✗ | ✓ | ✓ | ✗ | ✓ | ✗ | ✗ | ✓ |
| Convective Cell Change | ✗ | ✓ | ✓ | ✗ | ✓ | ✗ | ✗ | ✓ |
| Intensity Change | ✗ | ✓ | ✓ | ✗ | ✓ | ✗ | ✗ | ✓ |
| Dynamic Consistency Performance | ✗ | ✓ | ✗ | ✓ | ✓ | ✗ | ✓ | ✓ |
| Move Direction | ✗ | ✓ | ✓ | ✗ | ✓ | ✗ | ✗ | ✓ |
| Speed Difference | ✗ | ✓ | ✗ | ✓ | ✓ | ✗ | ✗ | ✓ |
| Rotation Center | ✗ | ✓ | ✓ | ✗ | ✓ | ✗ | ✗ | ✓ |
| Difference in Generation | ✗ | ✓ | ✗ | ✓ | ✓ | ✗ | ✗ | ✓ |
| Difference in Dissipation | ✗ | ✓ | ✗ | ✓ | ✓ | ✗ | ✗ | ✓ |
| Sharpness Performance | ✓ | ✗ | ✗ | ✓ | ✗ | ✓ | ✓ | ✓ |
| Shape Type | ✗ | ✓ | ✓ | ✗ | ✓ | ✗ | ✗ | ✓ |
| Shape Mismatch Direction | ✗ | ✓ | ✗ | ✓ | ✓ | ✗ | ✗ | ✓ |
| Shape Mismatch Reason | ✗ | ✓ | ✗ | ✓ | ✓ | ✗ | ✗ | ✓ |
| Artifacts Direction | ✗ | ✓ | ✗ | ✓ | ✓ | ✗ | ✗ | ✓ |
| Organization Degree | ✗ | ✓ | ✓ | ✗ | ✓ | ✗ | ✗ | ✓ |
| Distribution | ✓ | ✗ | ✓ | ✗ | ✗ | ✓ | ✗ | ✓ |
| Overall Performance (Sequence) | ✗ | ✓ | ✗ | ✓ | ✓ | ✗ | ✓ | ✓ |
| Overall Performance (Frame) | ✓ | ✗ | ✗ | ✓ | ✓ | ✗ | ✓ | ✓ |

system between the prediction and the ground truth. (c) Rotation Center. The spatial location that acts as the center of rotation for convective system evolution.

**Convective Cycle**.
- **Genesis**. (a) Difference in Generation. The difference in the number of newly generated convective cells between the prediction and the ground truth over the entire sequence.
- **Dissipation**. (a) Difference in Dissipation. The difference in the number of dissipated convective cells between the prediction and the ground truth throughout the sequence.

**Morphology**.

- **Sharpness**. (a) Sharpness Performance. The degree of similarity between the fine-grained contours in the prediction and those in the ground truth.
- **Shape**. (b) Shape Type. The morphological pattern of the convective system in the observation. (c) Shape Mismatch Direction. The directions in which the evolution trend of the convective shape in the prediction diverges from that in the ground truth. (d) Shape Mismatch Reason. The underlying cause contributing to the mismatch in convective morphology between the prediction and observation. (e) Artifacts Direction. The directions in which artificial patterns appear in the predicted sequence that do not exist in the observation. (f) Organization Degree. The temporal trend of structural organization in the ground truth reflects how orderly the convective system is over time. (g) Distribution. The directional distribution of precipitation in the observation

An overview of the properties associated with each attribute is demonstrated in Tab. A1.

|  | +5min | +15min | +25min | +35min | +45min | +55min |

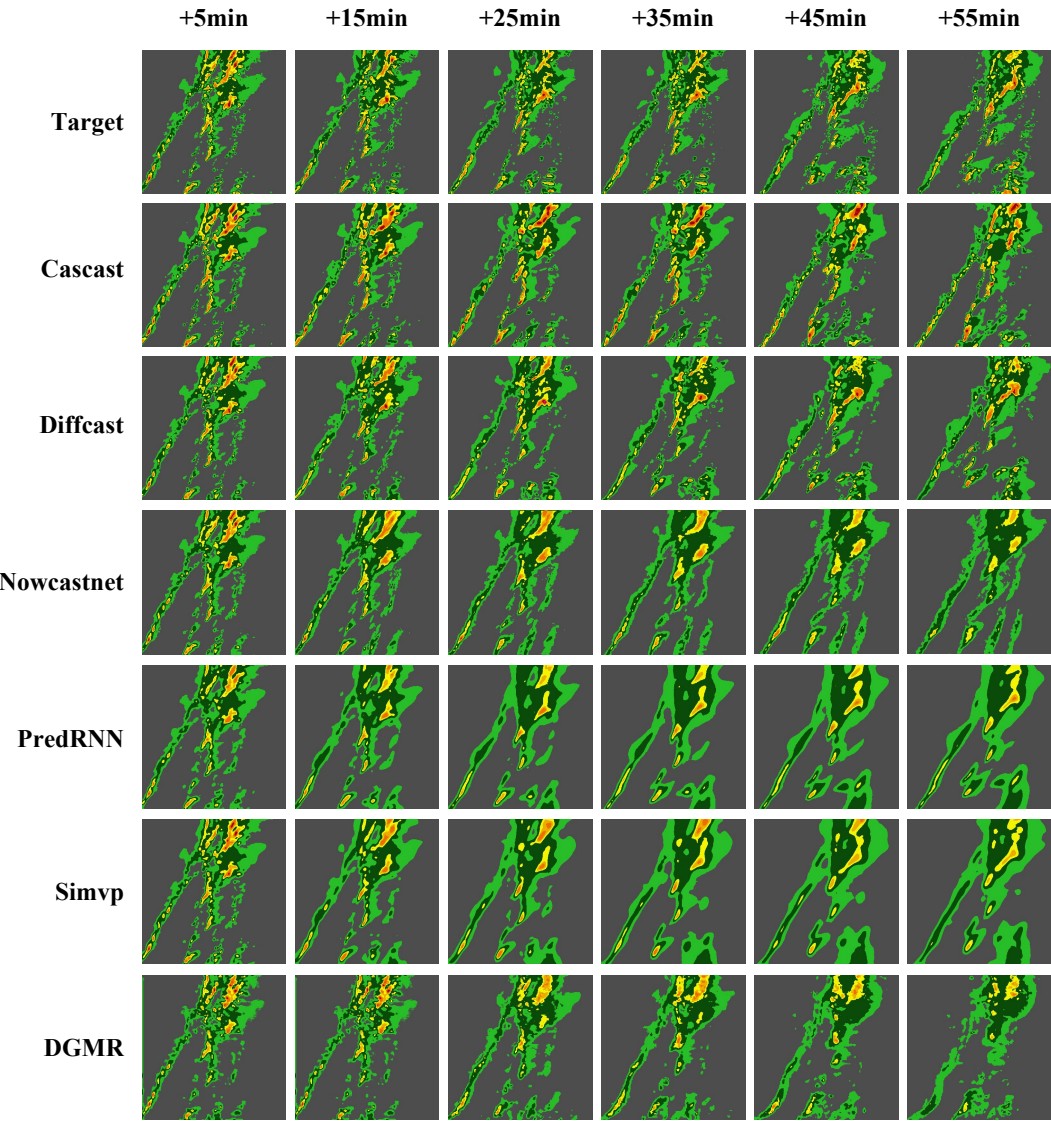

Figure A1: **A set of example forecasts** on SEVIR.

## B.2 Details of Raw Data Statistics

To ensure the diversity of samples in RawRQA-20K, we consider both a wide range of storm event types and a diverse set of generative models. First, our RawRQA-20K covers seven storm event types, including flash flood, flood, funnel cloud, hail, heavy rain, thunderstorm wind, and tornado. Due to their strong convective nature and high impact, these storm events pose significant challenges for forecasting and contribute to a diverse sample space. The number of samples for each event type is summarized in Tab. A2. The coverage area of storm events is shown in Fig. A2.

We employ a total of seven representative nowcasting models to generate prediction samples. As illustrated in Fig. A1, these models produce diverse samples that reflect a wide range of forecast qualities. For example, Cascast tends to over-predict in high-value regions, yet generally exhibits superior performance in detail reconstruction and dynamic consistency. In contrast, DGMR often introduces substantial artifacts, which significantly degrade the overall quality. Meanwhile, PredRNN suffers from severe temporal blurring and exhibits poor performance in "high value retain". These varied quality issues are reflected in the corresponding differences across the assessment reports.

Table A2: **Statistics** of RawRQA-20K.

| Event type | Flash flood | Flood | Funnel cloud | Hail | Heavy rain | Thunderstorm wind | Tornado |
|---|---|---|---|---|---|---|---|
| # of events | 218 | 121 | 58 | 556 | 55 | 1030 | 121 |

## B.3 Details of Human Annotation Questionnaire

For the human-annotated attributes listed in Tab. A1, we employed an annotation pipeline to ensure consistency and quality. First, for each attribute, we designed a corresponding multiple-choice question, with domain experts defining clear annotation guidelines. Second, a small set of pilot samples was used to evaluate annotation quality from several annotation companies. The company with the most accurate performance was selected for large-scale annotation. Third, all annotators underwent standardized training to align their understanding with expert standards. Each annotator completed a trial annotation set, which was reviewed by experts who provided feedback and corrected any misinterpretations. Fourth, upon completion of each annotation batch, a cross-validation step is conducted by different annotators to ensure quality. Finally, after annotation, domain experts performed quality control by randomly sampling and reviewing 35% of the samples in each batch. A batch would be accepted only if the sampled annotations met the quality standards; otherwise, the annotators were required to re-annotate the entire batch.

## B.4 Automated Generation

As shown in Tab. A1, 20 attributes are grounded in score-based metrics, where automated annotation provides more precise and consistent results compared to manual labeling. In this process, all the required thresholds or parameters are determined with the assistance of domain experts. The corresponding computation procedures for these attributes are detailed below.

- False Alarm Performance. First, we calculate the false alarm rate. Let $\mathcal{G}$ and $\mathcal{P}$ denote the sets of pixels with precipitation in the ground truth and the prediction, respectively. Define Hits as $H = |\mathcal{G} \cap \mathcal{P}|$ and False Alarms as $(F = |\mathcal{P} \setminus \mathcal{G}|)$. The false alarm rate is given by:

$$\text{false alarm rate} = \frac{F}{H + F} \tag{A1}$$

  Thresholds [0.1, 0.2, 0.3] are selected to categorize the false alarm rate into four performance levels("Great", "Good", "Fair", "Poor").

- Miss Performance. Similar to the false alarm rate, we compute the miss rate based on the binary masks. Following SEVIR, we define a pixel as having precipitation if its value exceeds 16. Let $\mathcal{G}$ and $\mathcal{P}$ denote the sets of pixels with precipitation in the ground truth and prediction. Define Hits as $H = |\mathcal{G} \cap \mathcal{P}|$ and Misses as $M = |\mathcal{G} \setminus \mathcal{P}|$. The miss rate is defined as:

$$\text{miss rate} = \frac{M}{H + M} \tag{A2}$$

  Thresholds[0.1, 0.2, 0.4] are used to categorize the miss rate into four performance levels.

- Sharpness Performance. Following SRViT, we evaluate the sharpness of the prediction and the ground truth using the Sobel filter. Specifically, let $S_{gt}$ and $S_{pred}$ denote the mean Sobel value of the ground truth and the prediction, respectively:

$$S_{gt} = \frac{1}{N} \sum_{i=1}^{n} \text{Sobel}(\text{gt})_i, \ S_{pred} = \frac{1}{N} \sum_{i=1}^{n} \text{Sobel}(\text{Pred})_i \tag{A3}$$

  We then compute the relative difference:

$$d = \begin{cases} 2 - \left| \frac{S_{\text{pred}}}{S_{\text{gt}}} \right|, & \text{if } \left| \frac{S_{\text{pred}}}{S_{\text{gt}}} \right| > 1 \\ \left| \frac{S_{\text{pred}}}{S_{\text{gt}}} \right|, & \text{otherwise} \end{cases} \tag{A4}$$

  Finally, we clip negative values to zero, and define the sharpness score as:

$$\text{sharpness score} = \max(0, d) \tag{A5}$$

  Thresholds [0.5, 0.7, 0.9] are used to categorize the sharpness into four levels.

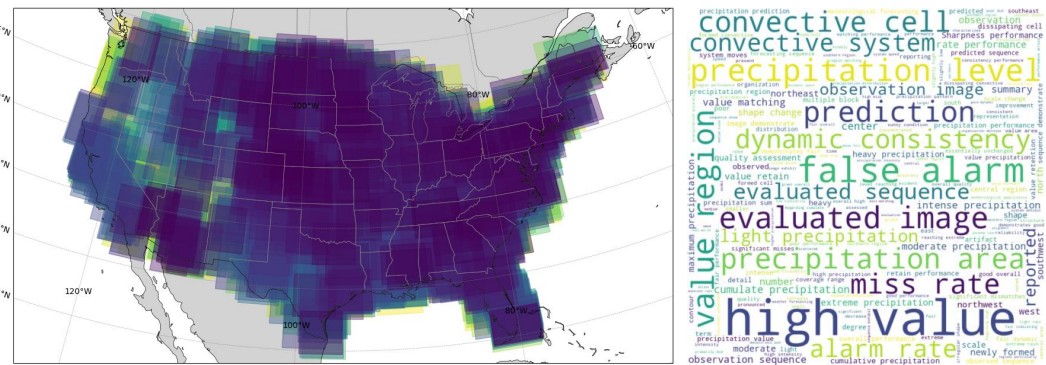

Figure A2: **Coverage area** of selected storm events in our RQA-70K dataset, which spans across the CONUS region.

Figure A3: **Wordcloud** map of our introduced RQA-70K dataset.

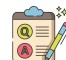 Annotation Questionnaire

### Caption

**1.** What is the moving direction of the convective system?
A. ↑     B. ↓     C. →     D. ←
E. ↘     F. ↖     G. ↗     H. ↙
I. ↻     J. ↺

**2.** How does the number of convective cells change?
A. increase     C. remain the same
B. decrease

**3.** How does the intensity of convective system change?
A. increase     C. remain the same
B. decrease

**4.** What is the rotate center of the convective system?
A. ↑     B. ↓     C. →     D. ←
E. ↘     F. ↖     G. ↗     H. ↙
I. center          J. no rotation

**5.** How does the coverage area of convective system change?
A. increase     C. remain the same
B. decrease

**6.** How does the organization degree of convective system change?
A. increase     C. remain the same
B. decrease

**7.** What is the shape of convective system?
A. scattered       F. multi-block-like
B. banded          G. multi-arc-shaped
C. block-like      H. multi-banded
D. large patch-like  I. spiral shaped
E. arc shaped      J. Irregular shaped

**8.** How does the shape of convective system change?
A. merge     E. split
B. stretch   F. disappear
C. shrink    G. form
D. dilate    H. remain the same

### Comparison

**1.** In which directions are the diff. in shape change most severe?
A. ↑     B. ↓     C. →     D. ←
E. ↘     F. ↖     G. ↗     H. ↙
I. center          J. remains the same

**2.** What is the main issue within the direct. with most diff. in shape change?
A. scale diff.     C. position diff.
B. diff. of convective cell numbers

**3.** What are the directions that have artifacts?
A. ↑     B. ↓     C. →     D. ←
E. ↘     F. ↖     G. ↗     H. ↙
I. center          J. remains the same

**4.** The scale of generated convective cell in the prediction is
A. larger     C. basically the same
B. smaller

**5.** The scale of dissipated convective cell in the prediction is
A. larger     C. basically the same
B. smaller

**6.** The movement speed of the convective cycle in the prediction is
A. faster     C. basically the same
B. slower

### Rating

**1.** What is the overall performance of the predicted sequence?
A. great     B. good
C. fair      D. poor

**2.** What is the dynamic consistency performance of the predicted sequence?
A. great     B. good
C. fair      D. poor

**3.** What is the overall performance of the predicted image?
A. great     B. good
C. fair      D. poor

Figure A4: **Human annotation questionnaire** for the 17 attributes that require manual labeling.

- High Value Mismatch Performance. We first count the number of high-value pixels in both the prediction and the ground truth(*i.e.*, pixels with intensity values greater than 219), denoted as $N_{pred}$ and $N_{gt}$, respectively. The relative error is computed as:

$$\mathcal{E}_{rel} = \left| \frac{N_{gt} - N_{pred}}{N_{gt}} \right| \tag{A6}$$

Table A3: Structure of **detailed descriptions** for each general attribute.

| General Attributes | Detailed Description |
| --- | --- |
| High Value Mismatch | In the *high value mismatch direction*, the prediction is *high value mismatch type* (over-predict / under-predict). |
| Miss | In the *Miss direction*, the *raw rainfall level* is misclassified as *miss rainfall level*. |
| Cumulate Precipitation | In the *mismatch direction*, the cumulate precipitation is *cumulate precipitation difference*. |
| High Value Retain | In the *high value mismatch direction*, the prediction is *high value mismatch type* (over-predict / under-predict). |

The high value mismatch score is subsequently defined as

$$\text{high value mismatch score} = \min(1, \max(0, 1 - \mathcal{E})) \tag{A7}$$

Thresholds [0.3, 0.6, 0.8] are used to categorize the high value mismatch into four levels.

- High Value Retain Performance. The high-value retain score is computed as the average high-value mismatch score across all frames. The same thresholds [0.3, 0.6, 0.8] are used to categorize the performance into four levels.

- Cumulate Precipitation Performance. First, we compute the total precipitation in the prediction and ground truth, denoted as $P_{pred}$ and $P_{gt}$, respectively. We then calculate the relative precipitation error and define the cumulate precipitation score using the same method as in the computation of $\mathcal{E}_{\text{rel}}$ and the high-value mismatch score. Thresholds [0.93, 0.97, 0.99] are applied to categorize the performance levels.

To provide a detailed characterization of the general attributes, we divide each image into a $3 \times 3$ grid, resulting in nine spatial regions corresponding to nine directional sectors. For each general attribute, its detailed description is formulated as a combination of directional information and the associated prediction issue. For example, in the case of false alarms, a typical description takes the form of "in the *FAR direction*, the *raw rainfall level* is false alarmed as the *FAR rainfall level*." This expression involves three distinct attributes, whose construction is detailed below.

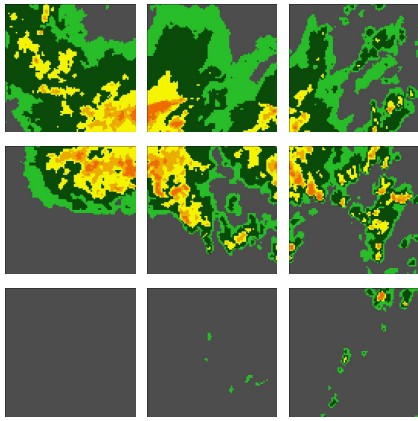

Figure A5: **Gridding** of the image into $3 \times 3$ patches, each representing a directional sector.

*Raw Rainfall Level.* First, we compute the number of missed pixels for each rainfall intensity level. To incorporate the varying importance of different rainfall levels, we align with domain experts and assign weights [1, 1.5, 2.5, 5, 10, 20], corresponding to increasing rainfall intensity from "light" to "extreme". Higher rainfall levels are given greater emphasis. We then compute the weighted sum of missed pixels for each level, ranking them in descending order, and identify the rainfall level with the highest weighted missing pixel count.

*FAR Rainfall Level.* For each raw rainfall level, we examine the corresponding locations in the prediction and count the occurrences of each predicted rainfall level. The rainfall level with the highest pixel count that is lighter than the raw rainfall level is selected as the FAR rainfall level.

*FAR Direction.* For each raw rainfall level, we compute the false alarm rate across different directions. We also count the number of pixels with raw rainfall level in each direction. To ensure both a high false alarm rate and a large false alarm area, We sort the directions by false alarm rate in descending order, and restrict our selection to those whose raw rainfall level pixel counts are among the top two. The first direction satisfying this condition is selected as the FAR direction.

For other general attributes, the structure of their detailed descriptions is summarized in Tab. A3, and the construction of their underlying attributes follows a similar procedure as in FAR.

**Bias from the usage of LLM**. We use GPT-4o to organize annotated attributes into assessment descriptions, which may introduce potential bias, including:

*Style bias*. The structure of the reports may be overly uniform and fail to reflect expert diversity.

*Accuracy bias*. The generated content does not always align with the visual information.d

*Redundacy bias*. The presence of unnecessary information may reduce clarity.

*Attribute Omission bias*. Less prominent yet important features may be overlooked.

# C    More Results

Table A4: **Few-shot Results** on general attributes for the frame rating and frame assessment tasks. Accuracy is used as the metric for the frame rating task. RadarQA surpasses all methods.

| Methods | | Frame Rating | | | | | Frame Assessment | | | | |
|---|---|---|---|---|---|---|---|---|---|---|---|
| | | Overall | False Alarm | Miss | High Value | Sharpness | BLEU | BERTScore | ROUGE-L | METEOR | GPT4Score |
| one shot | GPT4o | 43.72 | 26.40 | 29.65 | 27.09 | 49.19 | 0.164 | 0.782 | 0.448 | 0.372 | 5.33 |
| | Claude3.7 sonnet | 37.79 | 35.00 | 25.00 | 25.47 | 45.58 | 0.136 | 0.773 | 0.416 | 0.371 | 5.39 |
| | Gemini2.5 pro | 30.70 | 29.88 | 30.93 | 34.07 | 42.44 | 0.102 | 0.748 | 0.368 | 0.355 | 6.01 |
| Three shot | GPT4o | 52.79 | 32.67 | 33.60 | 29.65 | 52.21 | 0.167 | 0.787 | 0.456 | 0.383 | 5.31 |
| | Claude3.7 sonnet | 28.49 | 32.09 | 13.60 | 24.53 | 26.98 | 0.158 | 0.786 | 0.440 | 0.389 | 5.11 |
| | Gemini2.5 pro | 33.72 | 29.19 | 32.32 | 35.81 | 44.41 | 0.140 | 0.767 | 0.410 | 0.364 | 5.45 |
| Ours | RadarQA | **61.51** | **65.35** | **67.67** | **69.19** | **78.60** | **0.213** | **0.809** | **0.512** | **0.420** | **6.87** |

Table A5: More results on ablation studies of multi-stage training strategy on **rating tasks**.

| Stage-1 | Stage-2 | Stage-3 | Frame | | | | | Sequence | | | |
|---|---|---|---|---|---|---|---|---|---|---|---|
| | | | Overall | False Alarm | Miss | High Value Mismatch | Sharpness | Overall | Dynamic Consistency | Cumulate Precipitation | High Value Retain |
| ✗ | ✗ | ✗ | 20.10 | 36.40 | 30.00 | 16.51 | 35.93 | 7.99 | 16.10 | 17.49 | 23.22 |
| ✓ | ✗ | ✗ | 60.93 | 63.37 | 61.63 | 63.02 | 71.28 | 61.42 | 42.44 | 42.20 | 74.53 |
| ✓ | ✓ | ✗ | 59.77 | **68.14** | **67.67** | 65.00 | 74.19 | 61.55 | **64.17** | 42.82 | 77.78 |
| ✓ | ✗ | ✓ | 61.28 | 65.00 | 66.40 | **69.88** | 78.14 | 65.42 | 52.31 | **49.44** | **81.52** |
| ✓ | ✓ | ✓ | **61.51** | 65.35 | 67.67 | 69.19 | **78.60** | 66.17 | 53.31 | 48.94 | 80.52 |

Table A6: More results on ablation studies of multi-stage training strategy on **assessment tasks**.

| Stage-1 | Stage-2 | Stage-3 | Frame | | | | | Sequence | | | | |
|---|---|---|---|---|---|---|---|---|---|---|---|---|
| | | | BLEU | BERTScore | ROUGE_L | METEOR | GPT-4 Score | BLEU | BERTScore | ROUGE_L | METEOR | GPT-4 Score |
| ✗ | ✗ | ✗ | 0.122 | 0.75 | 0.389 | 0.332 | 3.81 | 0.09 | 0.745 | 0.281 | 0.342 | 3.92 |
| ✓ | ✗ | ✗ | 0.195 | 0.799 | 0.498 | 0.417 | 6.40 | **0.212** | 0.812 | 0.429 | 0.453 | 6.22 |
| ✓ | ✗ | ✓ | 0.212 | **0.810** | 0.511 | 0.423 | 6.83 | 0.211 | **0.816** | 0.431 | 0.461 | 6.56 |
| ✓ | ✓ | ✓ | **0.213** | 0.809 | **0.512** | **0.420** | **6.87** | 0.212 | 0.815 | 0.436 | 0.461 | 6.58 |

Table A7: **Comparison** with traditional weather analysis and general IQA methods on frame rating task. The threshold used for weather-related metrics is 74.

| Methods | Weather ralated metrics | | | | | | IQA methods | | Ours |
|---|---|---|---|---|---|---|---|---|---|
| | CSI | POD | FAR | Bias | ACC | ETS | DISTS | LPIPS | RadarQA |
| Accuracy | 41.74 | 42.79 | 39.07 | 39.42 | 39.53 | 43.60 | 53.60 | 46.63 | **61.51** |
| SRCC | 0.26 | 0.28 | 0.15 | 0.23 | 0.21 | 0.29 | 0.55 | 0.39 | **0.62** |
| PLCC | 0.27 | 0.29 | 0.16 | 0.20 | 0.22 | 0.29 | 0.56 | 0.43 | **0.64** |

Table A8: **Ablation studies of different model sizes**. Frame / sequence rating tasks are evaluated in average accuracy, while frame / sequence assessment tasks are assessed in GPT-4 Score.

| Model size | Rating | Assessment |
|---|---|---|
| 3B | 63.44 / 59.71 | 6.77 / 6.36 |
| 7B | 68.46 / 62.24 | 6.87 / 6.58 |

**Few-shot evaluation on frame rating task and frame assessment task**. We further evaluate the performance of different API-based models. As shown in Tab. A4, although other models are evaluated under few-shot settings, RadarQA consistently outperform all baselines without requiring any additional examples, demonstrating the effectiveness of RadarQA.

**Ablation studies on multi-stage training strategy**. For our multi-stage training strategy, we further examine the effectiveness of each stage across different metrics, as shown in Tab. A5 and Tab. A6. First, applying reinforcement learning significantly improves performance on reasoning-related metrics such as false alarm and miss rates. After supervised fine-tuning, the model leverages its ability on interpreting learned from assessment tasks to better rate general attributes. Finally, the full training strategy achieves the best performance on most metrics.

**Comparison with domain-specific baselines**. We compare RadarQA with weather-related metrics and general IQA methods. We use accuracy, PLCC, and SRCC as the evaluation metrics, which reflect the consistency between the evaluation results and the expert annotations. As shown in Tab. A3, RadarQA significantly outperforms the baselines across all three metrics.

**Ablation studies on different model sizes**. We further evaluate the performance of different model sizes under the same training strategy using Qwen-2.5-VL series. As shown in Tab. A8, the 3B model shows a slight drop in performance while using fewer parameters.

**Qualitative results**. More qualitative results of assessment tasks are shown in Fig. A6 and Fig. A7.

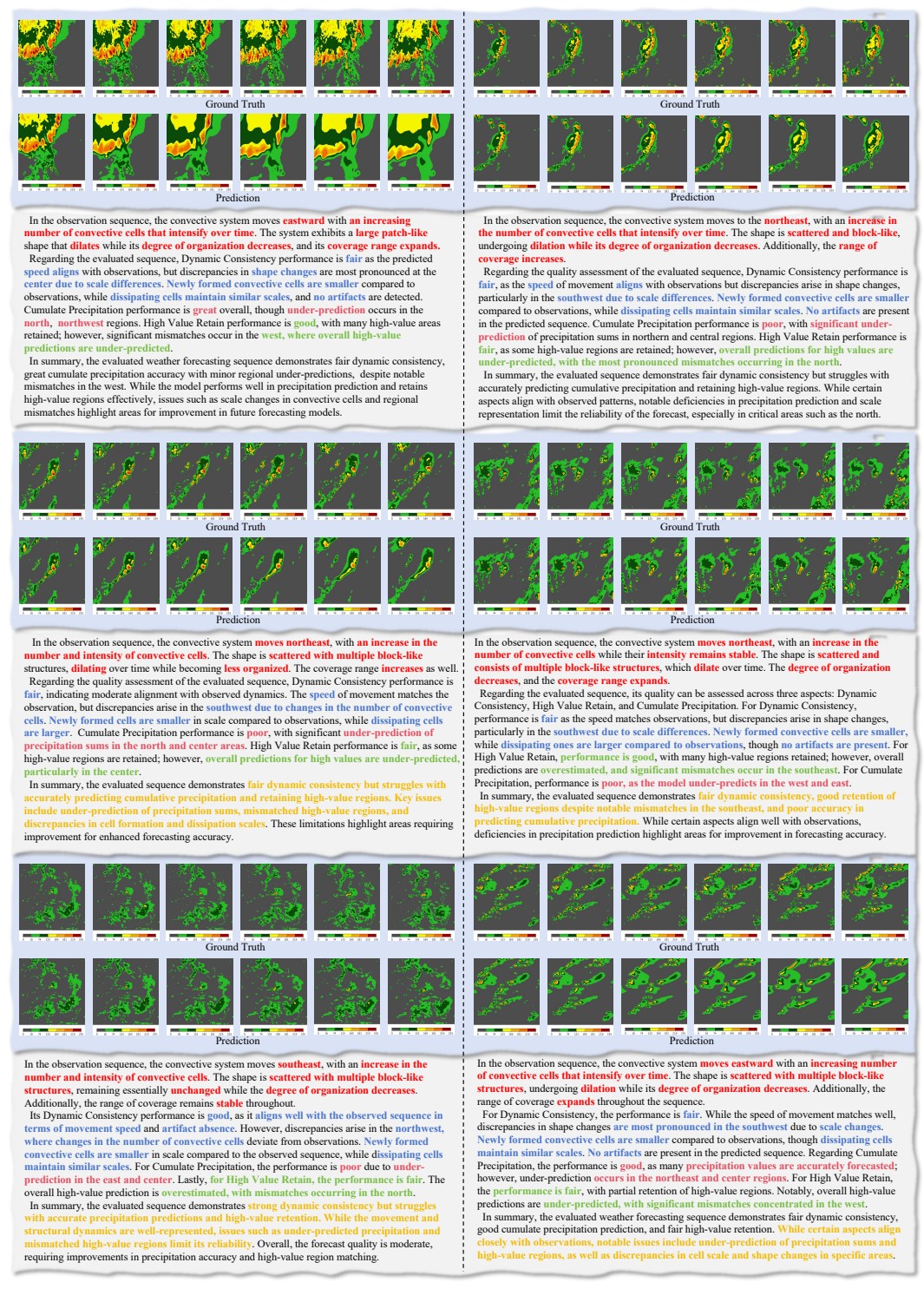

In the observation sequence, the convective system moves **eastward** with **an increasing number of convective cells that intensify over time**. The system exhibits a **large patch-like** shape that **dilates** while its **degree of organization decreases**, and its **coverage range expands**.
Regarding the evaluated sequence, Dynamic Consistency performance is **fair** as the predicted **speed aligns** with observations, but discrepancies in **shape changes** are most pronounced at the **center due to scale differences**. **Newly formed convective cells are smaller** compared to observations, while **dissipating cells maintain similar scales**, and **no artifacts** are detected. Cumulate Precipitation performance is **great** overall, though **under-prediction** occurs in the **north**, **northwest** regions. High Value Retain performance is **good**, with many high-value areas retained; however, significant mismatches occur in the **west, where overall high-value predictions are under-predicted**.
In summary, the evaluated weather forecasting sequence demonstrates fair dynamic consistency, great cumulate precipitation accuracy with minor regional under-predictions, despite notable mismatches in the west. While the model performs well in precipitation prediction and retains high-value regions effectively, issues such as scale changes in convective cells and regional mismatches highlight areas for improvement in future forecasting models.

In the observation sequence, the convective system moves to the **northeast**, with an **increase in the number of convective cells that intensify over time**. The shape is **scattered and block-like**, undergoing **dilation while its degree of organization decreases**. Additionally, the **range of coverage increases**.
Regarding the quality assessment of the evaluated sequence, Dynamic Consistency performance is **fair**, as the **speed** of movement **aligns** with observations but discrepancies arise in shape changes, particularly in the **southwest due to scale differences**. **Newly formed convective cells are smaller** compared to observations, while **dissipating cells maintain similar scales**. **No artifacts** are present in the predicted sequence. Cumulate Precipitation performance is **poor**, with **significant under-prediction** of precipitation sums in northern and central regions. High Value Retain performance is **fair**, as some high-value regions are retained; however, **overall predictions for high values are under-predicted, with the most pronounced mismatches occurring in the north**.
In summary, the evaluated sequence demonstrates fair dynamic consistency but struggles with accurately predicting cumulative precipitation and retaining high-value regions. While certain aspects align with observed patterns, notable deficiencies in precipitation prediction and scale representation limit the reliability of the forecast, especially in critical areas such as the north.

In the observation sequence, the convective system **moves northeast**, with **an increase in the number and intensity of convective cells**. The shape is **scattered with multiple block-like** structures, **dilating** over time while becoming **less organized**. The coverage range **increases** as well.
Regarding the quality assessment of the evaluated sequence, Dynamic Consistency performance is **fair**, indicating moderate alignment with observed dynamics. The **speed** of movement matches the observation, but discrepancies arise in the **southwest due to changes in the number of convective cells**. **Newly formed cells are smaller** in scale compared to observations, while **dissipating cells are larger**. Cumulate Precipitation performance is **poor**, with significant **under-prediction of precipitation sums in the north and center areas**. High Value Retain performance is **fair**, as some high-value regions are retained; however, **overall predictions for high values are under-predicted, particularly in the center**.
In summary, the evaluated sequence demonstrates **fair dynamic consistency but struggles with accurately predicting cumulative precipitation and retaining high-value regions. Key issues include under-prediction of precipitation sums, mismatched high-value regions, and discrepancies in cell formation and dissipation scales**. These limitations highlight areas requiring improvement for enhanced forecasting accuracy.

In the observation sequence, the convective system **moves northeast**, with an **increase in the number of convective cells** while their **intensity remains stable**. The shape is **scattered and consists of multiple block-like structures**, which **dilate** over time. The **degree of organization decreases**, and the **coverage range expands**.
Regarding the evaluated sequence, its quality can be assessed across three aspects: Dynamic Consistency, High Value Retain, and Cumulate Precipitation. For Dynamic Consistency, performance is **fair** as the speed matches observations, but discrepancies arise in shape changes, particularly in the **southwest due to scale differences**. **Newly formed convective cells are smaller,** while **dissipating ones are larger compared to observations**, though **no artifacts are present**. For High Value Retain, **performance is good**, with many high-value regions retained; however, overall predictions are **overestimated, and significant mismatches occur in the southeast**. For Cumulate Precipitation, performance is **poor, as the model under-predicts in the west and east**.
In summary, the evaluated sequence demonstrates **fair dynamic consistency, good retention of high-value regions despite notable mismatches in the southeast, and poor accuracy in predicting cumulative precipitation**. While certain aspects align well with observations, deficiencies in precipitation prediction highlight areas for improvement in forecasting accuracy.

In the observation sequence, the convective system moves **southeast**, with an **increase in the number and intensity of convective cells**. The shape is **scattered with multiple block-like structures**, remaining essentially **unchanged** while the **degree of organization decreases**. Additionally, the range of coverage remains **stable** throughout.
Its Dynamic Consistency performance is **good**, as it **aligns well with the observed sequence in terms of movement speed** and **artifact absence**. However, discrepancies arise in the **northwest, where changes in the number of convective cells** deviate from observations. **Newly formed convective cells are smaller** in scale compared to the observed sequence, while **dissipating cells maintain similar scales**. For Cumulate Precipitation, the performance is **poor** due to **under-prediction in the east and center**. Lastly, **for High Value Retain, the performance is fair**. The overall high-value prediction is **overestimated, with mismatches occurring in the north**.
In summary, the evaluated sequence demonstrates **strong dynamic consistency but struggles with accurate precipitation predictions and high-value retention. While the movement and structural dynamics are well-represented, issues such as under-predicted precipitation and mismatched high-value regions limit its reliability**. Overall, the forecast quality is moderate, requiring improvements in precipitation accuracy and high-value region matching.

In the observation sequence, the convective system **moves eastward** with an **increasing number of convective cells that intensify over time**. The shape is **scattered with multiple block-like structures**, undergoing **dilation** while its **degree of organization decreases**. Additionally, the range of coverage **expands** throughout the sequence.
For Dynamic Consistency, the performance is **fair**. While the speed of movement matches well, discrepancies in shape changes **are most pronounced in the southwest** due to **scale changes**. **Newly formed convective cells are smaller** compared to observations, though **dissipating cells maintain similar scales**. **No artifacts** are present in the predicted sequence. Regarding Cumulate Precipitation, the performance is **good**, as many **precipitation values are accurately forecasted**; however, under-prediction **occurs in the northeast and center regions**. For High Value Retain, the **performance is fair**, with partial retention of high-value regions. Notably, overall high-value predictions are **under-predicted, with significant mismatches concentrated in the west**.
In summary, the evaluated weather forecasting sequence demonstrates fair dynamic consistency, good cumulate precipitation prediction, and fair high-value retention. **While certain aspects align closely with observations, notable issues include under-prediction of precipitation sums and high-value regions, as well as discrepancies in cell scale and shape changes in specific areas**.

Figure A6: **Qualitative results** on sequence assessment task.

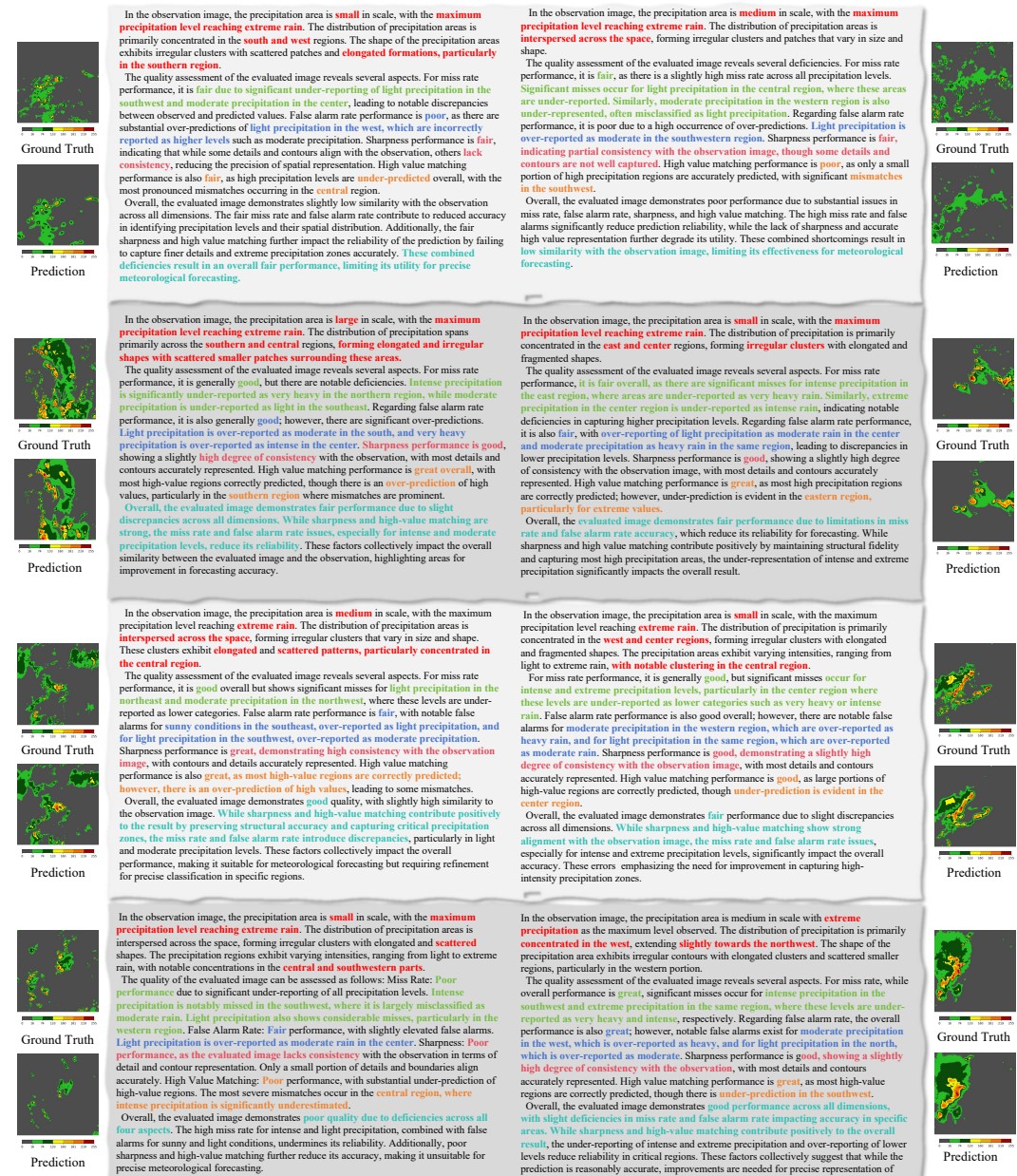

Figure A7: **Qualitative results** on frame assessment task.

Table A9: Question pool of *rating* task..

| # | Question |
|---|---|
| 1 | Could you score the prediction based on ${dim1}, ${dim2}, ${dim3}, and ${dim4}, and then provide an overall performance level? |
| 2 | Please assign levels to the prediction based on the four dimensions:${dim1}, ${dim2}, ${dim3}, and ${dim4}, and give an overall performance level. |
| 3 | How would you score the quality of the prediction on the dimensions of ${dim1}, ${dim2}, ${dim3}, and ${dim4}, and what would the overall level be? |
| 4 | Can you score the prediction using the four criteria: ${dim1}, ${dim2}, ${dim3}, and ${dim4}, and then provide an overall level? |
| 5 | Could you evaluate and score the prediction using ${dim1}, ${dim2}, ${dim3}, and ${dim4}, then provide a final overall performance level? |
| 6 | How would you score the prediction across dimensions of ${dim1}, ${dim2}, ${dim3}, and ${dim4}, and what would be the overall score? |
| 7 | Please score the prediction based on ${dim1}, ${dim2}, ${dim3}, and ${dim4}, then provide the overall performance level. |
| 8 | Could you score the prediction on ${dim1}, ${dim2}, ${dim3}, and ${dim4}, and then give an overall evaluation score for the prediction? |
| 9 | How would you rate the prediction across the four dimensions, ${dim1}, ${dim2}, ${dim3}, and ${dim4}, and what is the overall performance level? |
| 10 | How would you rate the prediction on the four dimensions, ${dim1}, ${dim2}, ${dim3}, and ${dim4}, and provide an overall performance level? |

Table A10: Question pool of *assessment* task.

| # | Question |
|---|---|
| 1 | Please start by describing the content of the observation, and then evaluate the quality of the prediction based on ${dim1}, ${dim2}, ${dim3}, and ${dim4}. Provide a comprehensive quality assessment report based on the 2 subtasks with a summary. |
| 2 | How would you describe the observation? Following that, could you evaluate the quality of the prediction across ${dim1}, ${dim2}, ${dim3}, and ${dim4}, then give a summary? |
| 3 | Provide a detailed quality report of the prediction. First describe the content of the observation, then focus on ${dim1}, ${dim2}, ${dim3}, and ${dim4} performance of the prediction. |
| 4 | Could you describe the observation's content, then assess the quality of the prediction according to ${dim1}, ${dim2}, ${dim3}, and ${dim4} in the format of a detailed report with summary? |
| 5 | Give a report of the prediction. First describe the content of the observation, then focus on ${dim1}, ${dim2}, ${dim3}, and ${dim4} of prediction. Finally, summarize your analysis. |
| 6 | Please describe the observation's content. Then, how would you assess the quality of the prediction based on ${dim1}, ${dim2}, ${dim3}, and ${dim4}? Give a detailed report with a summary. |
| 7 | What is your description of the observation? Afterward, could you evaluate the quality of the prediction on ${dim1}, ${dim2}, ${dim3}, and ${dim4}? Please provide a detailed report with a summary. |
| 8 | Start by describing the content of the observation, then assess the prediction on ${dim1}, ${dim2}, ${dim3}, and ${dim4}. Provide a detailed report with a summary. |
| 9 | How would you describe the content of the observation? Then, how would you evaluate the quality of the prediction on ${dim1}, ${dim2}, ${dim3}, and ${dim4}, and summarize your findings? Give a detailed report with a summary. |
| 10 | What content description would you give for the observation? Then, how would you evaluate the quality of the prediction across ${dim1}, ${dim2}, ${dim3}, and ${dim4}? Provide a detailed final report with a summary. |

