# OpenReview forum: "RadarQA: Multi-modal Quality Analysis of Weather Radar Forecasts"
_NeurIPS.cc/2025/Conference — NeurIPS 2025 poster_

### Official Review · Reviewer_6kbe · 2025-07-02

**Clarity:** 3
**Significance:** 3
**Originality:** 2
**Rating:** 5
**Confidence:** 3

**Summary:**

this paper proposes RadarQA, a multi-modal LLM system to judge quality of weather-radar forecasts; they make new 4-task paradigm (frame / sequence rating + assessment), build big RQA-70K data with human-and-script labels, and train model in three stage (SFT + GRPO RL + post-finetune). extensive experiments show large gain over open-source and closed models, even on out-of-distribution radar-SR task.

I really like how you combine the radar physics understanding with language evaluation, it gives a richer insight than just pure numbers,, it feels very helpful to meteorologists and I think it’s a big step forward and quite encouraging.


borderline accept: I feel the benefits slightly outweigh the concerns, so I encourage accepting..

**Questions:**

maybe add a concise, reader-friendly summary and reduce jargon and unpack acronyms

also maybe acknowledge that the underlying radar‐to‐forecast architecture builds on established vision-LLM methods, but emphasize your unique adaptations (e.g. handling of spatiotemporal radar echoes, specialized preprocessing)?

**Ethical Concerns:**

["NO or VERY MINOR ethics concerns only"]

**Final Justification:**

I appreciate the rebuttal from the authors and my main points are addressed with some satisfaction. Therefore, raising the score.

**Limitations:**

Yes

**Quality:**

3

**Strengths And Weaknesses:**

* The four-task paradigm you propose covers both static frames and dynamic sequences and gives both ratings and narrative, it’s kind of complete and well thought out, making the system versatile for different needs, and that design is very neat.
* Building the RQA-70K dataset with around seventy thousand samples labeled on fifteen frame and twenty-two sequence attributes is an impressive effort, and using a mix of human annotation and scripts makes it both efficient and reliable, so nice work on that.
* Training in three stages with fine-tuning, RL with GPT-based rewards, and post-fine-tuning to polish the style shows very good engineering, and the ablation studies clearly show each stage brings improvement, which is very convincing.
* Seeing the jump from around sixty-one to sixty-six percent accuracy on rating tasks and getting a GPT-4 score of 6.9 compared to 5.3 for other models is quite strong, and the out-of-distribution robustness even on super-resolution tasks is really promising, great results!
* Noting that the assessment generation also uses GPT-4 labels raises a little concern about possible bias and cost, but it still gives consistent high-quality text, so maybe just discuss that more in limitations.
* It would be even stronger to see more human evaluations rather than just GPT-4 scoring to avoid inflating the results, you know, some blind human review would add more confidence and make it feel more solid.
* The reliance on proprietary models and heavy compute, like eight A800 GPUs and generating twenty thousand storms, might be a barrier for smaller labs, so maybe you could suggest lighter versions or share code to help reproducibility, which would be awesome.
* The paper is detailed and sometimes dense with acronyms and appendix details, so reading can feel a bit tiring; maybe adding a clearer summary or reducing jargon would make it more accessible to broader audience.
* While the idea nicely applies MLLM techniques to weather radar, the core algorithm isn’t brand-new, it’s more an application than a fundamental breakthrough, but it’s still valuable and useful, so good job on practical impact.
* It might help to compare against stronger domain-specific baselines, like specialized CNN or quality control networks, to show even more advantage beyond general vision-LLMsthis would boost the significance.
* Missing an ethical and social impact discussion feels like a small gap because radar forecast assessments could influence public safety decisions, so a short caution section would be encouraging to see.

---

> ### Author Rebuttal · Authors · 2025-07-31
>
> We sincerely thank Reviewer 6kbe for the constructive comments and suggestions on RadarQA. We are glad that the completeness and versatility of our four-task paradigm was appreciated. We are encouraged by the reviewer's recognition of our strong performance in both in-distribution and out-of-distribution settings. We address the reviewer's comments in more detail below.
>
> ## Q1: Potential bias and cost of using GPT-4-generated labels.
>
> Thank you for the reviewer’s suggestion.
>
> First, the key attributes of our dataset is annotated by meteorological experts or calculated by meteorological metrics. GPT-4o is only used to organize these attributes into a fluent language. Therefore, the potential bias introduced by GPT-4o is very small.
>
> Second, the total cost of GPT-4o labeling was approximately $700. We will include a discussion of the limitations associated with GPT-4o annotations in the final version of the manuscript.
>
> ## Q2: Blind human review.
>
> We conduct human experts evaluation in the "Expert study" section (Page 9, Line 326-332) of the submission, where our method show clear advantage over GPT-4o. Specifically, we first randomized the ground truth, RadarQA’s outputs, and GPT-4o’s outputs. Then, we invite nine meteorology experts to participate in the expert study. Based on the observed images, model predictions, and different quality assessment reports, the experts considered the following three dimensions to provide an overall score: (1) Accuracy of the textual content, (2) Whether the issues of concern to the experts are mentioned in the report, (3) The density of effective information.
>
> ## Q3: Reproducibility challenges due to reliance on proprietary models and high computational cost.
>
> First, we will release all codes, collected datasets, and model weights for reproducibility.
>
> Second, we highlight that fine-tuning a 7B-parameter model is not so costly, and can be reliably performed on 2 NVIDIA RTX 4090 GPUs, with inference requiring only a single NVIDIA RTX 4090 GPU.
>
> Third, we also trained a smaller 3B model by using Qwen2.5-VL-3B-Instruct as the base model under the same training settings. The results are given below.
>
> | Accuracy                    | Overall | False Alarm | Miss   | High Value | Sharpness | Overall | Dynamic Consistency | Cumulate Precipitation | High Value Retain |
> |----------------------------|---------|-------------|--------|-------------|-----------|---------|----------------------|-------------------------|--------------------|
> |          | Frame|Frame|Frame|Frame|Frame|Sequence|Sequence|Sequence|Sequence|
> | Qwen2.5-VL-3B Instruct     | 60.93   | 58.95       | 57.33  | 65.81       | 74.19     | 64.42   | 53.43                | 46.19                   | 74.78              |
> | Qwen2.5-VL-7B Instruct     | 61.51   | 65.35       | 67.67  | 69.19       | 78.60     | 66.17   | 53.31                | 48.94                   | 80.52              |
>
> | Frame / Sequence               | BERTScore     | BLEU          | METEOR        | ROUGE_L       | GPT4Score   |
> | ---------------------- | ------------- | ------------- | ------------- | ------------- | ----------- |
> | Qwen2.5-VL-3B Instruct |0.811 / 0.815       |0.212 / 0.216       |0.425 / 0.465       |0.513 / 0.434       |6.77 / 6.36      |
> | Qwen2.5-VL-7B Instruct | 0.809 / 0.815 | 0.213 / 0.212 | 0.420 / 0.461 | 0.512 / 0.436 | 6.87 / 6.58 |
>
>
> ## Q4: Comparison with stronger domain-specific baselines.
>
> We compare RadarQA with domain specific baselines in the table below. The results show that RadarQA significantly outperforms all selected baselines, highlighting its effectiveness in aligning with human experts.
>
> The domain specific baselines are:  **Weather specific metrics** including CSI, POD, FAR, bias, and ETS, with thresholds set at 74 and 160, and 181, and **General FR-IQA methods** including PSNR, SSIM, LPIPS and DISTS.
>
>
> | Metric       | Accuracy | PLCC | SRCC |
> |--------------|----------|------|------|
> | **Weather-related metrics** |          |      |      |
> | csi_74       | 41.74    | 0.27 | 0.26 |
> | pod_74       | 42.79    | 0.29 | 0.28 |
> | far_74       | 39.07    | 0.16 | 0.15 |
> | bias_74      | 39.42    | 0.20 | 0.23 |
> | acc_74       | 39.53    | 0.22 | 0.21 |
> | ets_74       | 43.60    | 0.29 | 0.29 |
> | csi_160      | 44.42    | 0.43 | 0.41 |
> | pod_160      | 45.70    | 0.46 | 0.44 |
> | far_160      | 38.60    | 0.10 | 0.12 |
> | bias_160     | 46.28    | 0.40 | 0.42 |
> | acc_160      | 38.02    | 0.06 | 0.05 |
> | ets_160      | 41.51    | 0.39 | 0.34 |
> | **IQA methods** |          |      |      |
> | psnr         | 41.05    | 0.28 | 0.25 |
> | ssim         | 42.44    | 0.33 | 0.32 |
> | lpips        | 46.63    | 0.43 | 0.39 |
> | dists        | 53.60    | 0.56 | 0.55 |
> | **Ours**     |          |      |      |
> | RadarQA      | 61.51    | 0.64 | 0.62 |
>
>
> ## Q5: Ethical and social impact discussion.
>
> We appreciate the suggestion. We acknowledge that weather prediction assessment may have a quite huge influence in humans' daily life and production. We will include a dedicated caution section addressing these considerations in the final version to ensure responsible use and awareness of the limitations of our work.

---

> > ### Comment · Reviewer_6kbe · 2025-08-05
> >
> > While I am not entirely satisfied with the first two points of the rebuttal, I think my main points have been addressed.
> >
> > Consider all these points, I raised my score from 4 to 4.5 (rounded to 5). I appreciate the efforts from the authors.

---

> ### Author Response · Authors · 2025-08-06
>
> We sincerely thank Reviewer 6kbe for recognizing our work. We will revise our manuscript carefully according to your suggestions. Below are more details about Q1 and Q2.
>
> ### Q1: Potential bias and cost of using GPT-4-generated labels.
>
> We use GPT-4o to organize annotated attributes into assessment descriptions for assessment tasks, which may introduce potential bias, including:
>
> 1. Style bias. The structure of generated reports might be similar can cannot reflect the expert diversity.
> 2. accuracy bias. The generated content may not fully consistent with the visual information in the label.
> 3. Redundancy bias. GPT-4o may generate descriptions that contain redundant information, which often reduces clarity of the evaluation.
> 4. Attribute Omission bias. GPT-4o may ignore less prominent yet important features, resulting in incomplete assessments.
>
> We will include a more detailed discussion of the potential biases introduced by GPT-4o in the Limitation section, covering issues such as style bias, accuracy bias, and others identified above.
>
> ### Q2: Blind human review.
>
> We appreciate the reviewer’s  insightful suggestion. We will carry out a larger-scale human validation and expert study to further support our findings, we are currently limited by time and thus unable to provide more accurate and comprehensive results at this stage.

---

### Official Review · Reviewer_Htbn · 2025-07-02

**Clarity:** 2
**Significance:** 3
**Originality:** 3
**Rating:** 4
**Confidence:** 3

**Summary:**

This paper introduces RadarQA, a multi-modal large language model-based method for quality analysis of weather radar forecasts. The authors argue that traditional score-based metrics (like CSI, POD, FAR) fall short of expert-level analysis in descriptive properties, interpretability, and miss the dynamic evolution of weather systems.

To address this, they propose: A new task definition with four sub-tasks spanning frame-level and sequence-level analysis, each with both rating (quantitative) and assessment (descriptive) components.

Then, the author proposed RQA-70K dataset constructed based on SEVIR weather data, multiple forecasting models, and a hybrid annotation pipeline combining human expert labels with automated metrics. RadarQA model trained via a three-stage pipeline: supervised fine-tuning, reinforcement learning with GRPO, and post-training. The experiment results demonstrating better performance over existing MLLMs, including GPT-4o and Claude-3.7.

**Questions:**

See in "Weakness".

**Ethical Concerns:**

["NO or VERY MINOR ethics concerns only"]

**Final Justification:**

Thanks for the author's detailed response.

The author's responses to Q1, Q2, and Q3 are convincing. The rebuttal addressed my concern.

I have updated my score to "Borderline Accept".

**Limitations:**

Yes. In "Conclusion and Limitation" section.

**Quality:**

3

**Strengths And Weaknesses:**

**Strengths:**

1.  The paper highlights a genuine gap between automated metrics and expert meteorological analysis. This is a crucial application scenario（AI in meteorology / AI for Science). The motivation is clearly articulated with concrete examples showing how descriptive analysis provides insights beyond numerical scores.

2. The four-task matrix (frame/sequence × rating/assessment) is thoughtfully designed to mirror how meteorologists evaluate forecasts.  The scientific attribute library incorporating physics-informed attributes (morphology, intensity, atmospheric physics properties) shows domain expertise and goes beyond simple metrics.

3. The key contribution of this paper is the RQA-70K dataset. The dataset construction process takes the hybrid annotation approach, combining human expertise with automated metrics. The use of GPT-4o for generating assessment reports from attributes is clever, addressing the challenge of obtaining consistent expert annotations. The dataset will benefit to the field.

**Weaknesses:**
1. While the paper compares against several MLLMs, it lacks comparison with traditional weather analysis methods or weather-specific ML models. A comparison with more traditional methods provides a better understanding of how MLLMs differ from previous methods. How much improvement has been made with the MLLM method, and what level of performance can be achieved by MLLMs.

2. The RL post-training part is not clear enough and lacks RL-specific analysis. What's your objective function or optimization goal? Just simple "Format Reward" and "Accuracy Reward" with a binary reward. But the structure output is a complex task, maybe this reward doesn't guide the model towards a better JSON structure. Accuracy reward doesn't consider attribute dependencies or correlations. Also, single epoch training may be insufficient for meaningful policy improvement.

3. Figure 2 is kind of a mess and hard to capture the key information. For example, what does each color represent? The author could provide a label in the figure.

---

> ### Author Rebuttal · Authors · 2025-07-31
>
> We sincerely thank Reviewer Htbn for detailed feedback. We greatly appreciate the recognition of our clearly motivated problem setting, dataset construction strategy. We are glad that the reviewer found our four-task matrix well aligned with real-world forecasting practices. Below, we address the reviewer's main concerns point by point.
>
>
> ## Q1: Comparison with traditional weather analysis methods and weather-specific ML models.
>
>
> First, we compare our method with meteorology-specific metrics and image quality assessment (IQA) metrics on the overall Frame Rating task in the table below. The accuracy, PLCC, and SRCC reflect the consistency between the evaluation results and the expert annotations. RadarQA significantly outperforms the baseline traditional metrics across all three evaluation metrics. While these traditional metrics are reasonable, there remains a noticeable gap between their outputs and expert judgment. These baseline metrics are meteorology-specific ones, including CSI, POD, FAR, bias, and ETS (with thresholds 74/160/181), and IQA metrics including PSNR, SSIM, LPIPS, and DISTS.
>
>
> | Metric       | Accuracy | PLCC | SRCC |
> |--------------|----------|------|------|
> | **Weather-related metrics** |          |      |      |
> | csi_74       | 41.74    | 0.27 | 0.26 |
> | pod_74       | 42.79    | 0.29 | 0.28 |
> | far_74       | 39.07    | 0.16 | 0.15 |
> | bias_74      | 39.42    | 0.20 | 0.23 |
> | acc_74       | 39.53    | 0.22 | 0.21 |
> | ets_74       | 43.60    | 0.29 | 0.29 |
> | csi_160      | 44.42    | 0.43 | 0.41 |
> | pod_160      | 45.70    | 0.46 | 0.44 |
> | far_160      | 38.60    | 0.10 | 0.12 |
> | bias_160     | 46.28    | 0.40 | 0.42 |
> | acc_160      | 38.02    | 0.06 | 0.05 |
> | ets_160      | 41.51    | 0.39 | 0.34 |
> | **IQA methods** |          |      |      |
> | psnr         | 41.05    | 0.28 | 0.25 |
> | ssim         | 42.44    | 0.33 | 0.32 |
> | lpips        | 46.63    | 0.43 | 0.39 |
> | dists        | 53.60    | 0.56 | 0.55 |
> | **Ours**     |          |      |      |
> | RadarQA      | 61.51    | 0.64 | 0.62 |
>
>
>
> Second, to the best of our knowledge, there are no existing works in the meteorological domain that apply machine learning models for quality evaluation. Traditionally, meteorological forecast quality assessment relies on two approaches: expert subjective judgment and objective metric-based scoring.
> 1. **Expert subjective judgment** requires high costs and specialized domain knowledge.
> 2. **Objective metric-based scoring** cannot comprehensively evaluate forecast results, lacking understanding and assessment of proprietary meteorological patterns (*e.g.*, convective systems).
>
> Therefore, we leverage the power of MLLMs to effectively integrate expert knowledge (text-based quality assessments) with objective metrics (level-based quality rating) for quality evaluation, enabling a comprehensive assessment that combines domain expertise and quantitative indicators at a lower cost.
>
> ## Q2: RL training details.
>
> We appreciate the reviewer's question. We have provided descriptions of the RL training details in the main paper (Page 7, Lines 258-269). We will elaborate on it below for completeness.
>
> **Overall RL method**. In the RL post-training stage, we adopt the GRPO algorithm to optimize the model's performance on the rating task. Given a query $q$, the model with old parameters $\pi_{old}$ generates $N$ different responses ${o_1, o_2, ..., o_N}$. Then, a custom reward model is used to compute the corresponding rewards ${r_1, r_2, ..., r_N}$. Unlike other online RL methods (e.g., PPO), GRPO utilizes group-wise averaging to estimate the relative advantage without requiring an additional value model. The computation is performed as follows:
>
> $$
> A_i=\frac{r_i-\text{mean}(\{r_1,r_2,...,r_N\})}{\text{std}(\{r_1,r_2,...,r_N\})}
> $$
>
> Based on the advantage function, the objective function can be expressed as:
>
> $$
> \mathcal{J}(\theta)=\mathbb{E} \left[ \min(\rho_i A_i,\ \text{clip}(\rho_i, \ 1 - \delta, \ 1 + \delta)\ A_i) - \beta \cdot D_{\mathbb{KL}}(\pi_{\theta_{new}}||\pi_{ref})\right]
> $$
>
> Here，$\rho_i=\frac{\pi_{\theta_{new}}(o_i|q)}{\pi_{\theta_{old}}(o_i|q)}$, $q\sim{Q}$ and $o_i\sim\pi_{\theta_{old}}$.
>
> **Format reward**. The format reward function uses regular expressions to determine whether the model's output can be correctly parsed. If the format of output can be successfully parsed into a JSON format and all keys exactly match those in the ground truth, the reward is 1; otherwise, the reward is 0.
>
> **Accuracy reward**. Let $n$ be the total number of keys in the ground truth (i.e. `n = len(gt_dict.keys())`). If the extracted answer has $k$ keys whose values match those in the ground truth, then the reward is $k/n$. Since RL training is only applied in rating tasks where the answer could be "poor", "fair", "good", and "great", it is easy to compare the predicted answer to the ground truth. The output is first parsed into a valid JSON format to extract the answer. If the parse fails, the reward is also 0.
>
> **Number of training epochs**. A shown in the table below, training with 1 epoch has led to good enough performance, and increasing the number of training epochs does not lead to significant performance improvements. Therefore, we choose to train for 1 epoch to balance computational cost and model performance.
>
> | Epochs |Overall | False Alarm | Miss | High Value | Sharpness | Overall | Dynamic Consistency | Cumulate Precipitation | High Value Retain |
> |--------|-------------|-------------|------|-------------|-----------|--------------|----------------------|-------------------------|--------------------|
> |         | Frame     | Frame     | Frame     | Frame     | Frame     | Sequence | Sequence | Sequence | Sequence |
> | 1      | 59.77       | 68.14       | 67.67 | 65.00       | 74.19     | 61.55        | 64.17                | 42.82                   | 77.78              |
> | 2      | 61.05       | 68.37       | 66.63 | 66.51       | 77.09     | 61.80        | 41.20                | 38.83                   | 77.40              |
> | 3      | 60.47       | 68.02       | 66.86 | 67.21       | 76.63     | 61.05        | 42.20                | 37.20                   | 76.40              |
>
>
> ## Q3: Labeling in Figure 2.
>
> We appreciate the reviewer's suggestion. In Figure 2, we use different colors to highlight some key information. We will simplify the use of colors to make it easier to read. We will also add a label to improve clarity and understanding.

---

> > ### Comment · Reviewer_Htbn · 2025-08-04
> >
> > Thanks for the author's detailed response.
> >
> > The author's responses to Q1, Q2, and Q3 are convincing. The rebuttal addressed my concern.
> >
> > I have updated my score to "Borderline Accept".

---

> > > ### Author Response · Authors · 2025-08-06
> > >
> > > Thank you very much for your valuable feedback and support! We are glad to hear that your concerns have been resolved. We will revise the paper carefully according to your suggestions. If you have any further comments or questions, please feel free to post them here for discussion.

---

### Official Review · Reviewer_AvwT · 2025-07-03

**Clarity:** 3
**Significance:** 3
**Originality:** 4
**Rating:** 5
**Confidence:** 3

**Summary:**

Thanks to the authors for their contribution. The paper introduces RadarQA, a Multi-modal Large Language Model (MLLM)-based system designed for quality analysis of weather radar forecasts. The proposed RadarQA employs visual and textual modalities to assess static (single frame) and temporal (sequence) radar predictions. The contributions can be summarized as follows-
* A task policy covering four evaluation types for static and sequence rating and assessment
* RQA-70k dataset annotated by a hybrid annotation pipeline combining human labeling with automated heuristics
* A multi-stage training pipeline for model improvement and comprehensive evaluation of the results for quality analysis in weather prediction

**Questions:**

My questions are inline below-

* The training strategy seems quite computationally intensive. I'm curious how this approach is reproducible across other weather associated tasks.
* The authors mention the importance of human annotation and how human annotation was used in curating the proposed RadarQA. However, the paper doesn't describe the process followed for human annotation. How were these annotators recruited? What were their demographics and background information? Ideally, this information is important for evaluating biases associated with data collection and curation.
* If the overall objective of the work is to perform quality analysis, I'm curious what the effect of using the radar outputs as images instead of raw values. It is observed that converting radar outputs to RGB or image format has an implicit loss of information. How is that being addressed in the proposed workflow?
* The authors are recommended to provide dataset samples and fine-tuning instruction prompts as a part of the text or appendix so that the work is reproducible.
* Training configuration for LoRA finetuning is missing
* I'm curious to see such low Bleu and Meteor scores. Is this justified for such a kind of evaluation, or is it due to custom outputs being evaluated? I missing be missing something here and would appreciate some insights here.

**Ethical Concerns:**

["NO or VERY MINOR ethics concerns only"]

**Final Justification:**

The authors have addressed the concerns in their rebuttals and explained quantitatively how the conversion of raw data to images is lossless. The authors are also committing to release their dataset and code publicly.

**Limitations:**

Yes. The authors have highlighted limitations of their work in the manuscript.

**Paper Formatting Concerns:**

A minor formatting issue: There's a typo in the center block- "Anotation" -> "Annotation" in the figure 4.

**Quality:**

3

**Strengths And Weaknesses:**

The proposed work addresses an important problem of assessing the quality of weather radar forecasts using MLLMs. I liked the task's policy formulation for including descriptive and domain-informed feedback. The training strategy is logical, and proper ablation studies are thoroughly described. The annotation pipeline includes human annotation, which is promising. Overall, the paper is well articulated.

Shortcomings

Since neither the proposed dataset nor the codes are openly available, the validation of the dataset and results is restricted. Additionally, the background for defining task attributes based on the scientific attributes library is not clear to me.

---

> ### Author Rebuttal · Authors · 2025-07-31
>
> We sincerely thank Reviewer AvwT and the area chair for the thoughtful feedback and appreciation for our training strategy and ablation studies. We also appreciate the positive comments on the annotation pipeline and clarity of the overall presentation. We will continue to refine the manuscript accordingly. Below, we address the reviewer's concerns in detail.
>
> ## Q1：Public availability of the proposed dataset and codes.
>
> We commit to releasing all datasets, codes, and model weights upon acceptance of the paper to support community usage and reproducibility.
>
>
> ## Q2：Background of task attribute definition based on the scientific attributes library.
>
> Our model is designed to evaluate the quality of short-term precipitation forecasting.
>
> Short-term precipitation forecasting [1, 2] is one of the most fundamental tasks in the meteorological domain. In our experiments, forecasting models are tasked with predicting the next 12 frames of precipitation based on the previous 10 frames. These predictions are intended to support early warning systems and meteorological agencies in making timely decisions to mitigate the impact of severe convective weather events.
>
> To evaluate the quality of such forecasts, meteorological experts typically rely on a combination of professional knowledge and formula-based indicators, aiming to ensure the reliability of assessments.
>
> Inspired by this evaluation paradigm, we collaborated with a frontline meteorologist with nearly 20 years of experience to construct a *scientific attribute library* for assessing forecast quality from both physical and perceptual standpoints, ensuring professionalism.
>
> Some attributes are derived from metrics, aligning with expert use of objective indicators, while others are designed based on perceptual and domain-specific meteorological knowledge, reflecting expert experience. This library allows for a comprehensive evaluation of model outputs that mirrors expert judgment. The detailed definitions and properties of all attributes can be found in the supplementary material.
>
> [1] Gong J, Bai L, Ye P, et al. Cascast: Skillful high-resolution precipitation nowcasting via cascaded modelling[J]. arXiv preprint arXiv:2402.04290, 2024.
>
> [2] Zhang Y, Long M, Chen K, et al. Skilful nowcasting of extreme precipitation with NowcastNet[J]. Nature, 2023, 619(7970): 526-532.
>
>
> ## Q3: Resource-intensive training strategy and reproducibility across diverse weather forecasting scenarios.
>
> We appreciate the reviewer's question. The proposed training strategy is indeed applicable to other weather-related tasks.
>
> First, fine-tuning a 7B-parameter model is not so costly, and can be reliably performed on two NVIDIA RTX 4090 GPUs, with inference requiring only a single GPU.
>
> Second, smaller models can also be used depending on the application scenario. We conducted experiments under the same setting using Qwen2.5-VL-3B-Instruct in the table below. Although there is a slight performance drop, it still demonstrates promising potential for quality assessment.
>
> | Accuracy                    | Overall | False Alarm | Miss   | High Value | Sharpness | Overall | Dynamic Consistency | Cumulate Precipitation | High Value Retain |
> |----------------------------|---------|-------------|--------|-------------|-----------|---------|----------------------|-------------------------|--------------------|
> |          | Frame|Frame|Frame|Frame|Frame|Sequence|Sequence|Sequence|Sequence|
> | Qwen2.5-VL-3B Instruct     | 60.93   | 58.95       | 57.33  | 65.81       | 74.19     | 64.42   | 53.43                | 46.19                   | 74.78              |
> | Qwen2.5-VL-7B Instruct     | 61.51   | 65.35       | 67.67  | 69.19       | 78.60     | 66.17   | 53.31                | 48.94                   | 80.52              |
>
>
> | Frame / Sequence               | BERTScore     | BLEU          | METEOR        | ROUGE_L       | GPT4Score   |
> | ---------------------- | ------------- | ------------- | ------------- | ------------- | ----------- |
> | Qwen2.5-VL-3B Instruct |0.811 / 0.815       |0.212 / 0.216       |0.425 / 0.465       |0.513 / 0.434       |6.77 / 6.36      |
> | Qwen2.5-VL-7B Instruct | 0.809 / 0.815 | 0.213 / 0.212 | 0.420 / 0.461 | 0.512 / 0.436 | 6.87 / 6.58 |
>
> Third, data is the key factor to adapt this MLLM-based method to diverse weather forecasting scenarios. Researchers can follow our proposed dataset construction pipeline to construct new datasets.
>
> Finally, with constructed datasets, the proposed training strategy is feasible for a wide range of tasks.
>
> ## Q4: Annotator profiles.
>
> First, we invited meteorological experts to define the annotation guidelines and standards.
>
> Second, we recruited a total of 16 annotators with undergraduate-level education or higher, all with backgrounds in meteorology and prior annotation experience.
>
> Then, annotators were asked to conduct a pilot annotation phase, during which their results were reviewed and validated by experts.
>
> Also, feedback from this phase was used to refine and clarify the annotation rules, aiming to minimize annotator bias and ensure consistency throughout the labeling process.
>
> ## Q5: Potential information loss.
>
> We appreciate the reviewer's concern.
>
> First, the conversion from raw radar data to RGB image is almost lossless. We need to convert the raw radar data (float32) into RGB format (uint8). Our statistics below with RMSE, MAE, R2, PSNR, and SSIM metrics show that this conversion is almost lossless.
>
> | RMSE  | MAE   | R2    | PSNR  | SSIM  |
> | ----- | ----- | ----- | ----- | ----- |
> | 0.001 | 0.0008 | 0.996 | 57.004 | 0.999 |
>
>
> Second, for better visualization and to facilitate understanding of different precipitation levels, we follow previous meterological works [1,2] to divide the precipitation into seven discrete levels (*i.e.*, sunny, light, moderate, heavy, very heavy, intense, and extreme), which is a common and standard visualization process in the meteorological field.
>
> [1] Gong J, Bai L, Ye P, et al. Cascast: Skillful high-resolution precipitation nowcasting via cascaded modelling[J]. arXiv preprint arXiv:2402.04290, 2024.
>
> [2] Robinson M, Evans J, Crowe B. En route weather depiction benefits of the NEXRAD vertically integrated liquid water product utilized by the corridor integrated weather system[C]//10th conference on aviation, range and aerospace meteorology, american meteorological society, portland, or. 2002.
>
>
> ## Q6：More details about the dataset samples and the fine-tuning instruction prompts.
>
> We commit to releasing all datasets, codes, and model weights upon acceptance of the paper to support community usage and reproducibility. The system prompt and instruction-tuning prompt used in our experiments will also be fully released.
>
> Some samples from RadarQA, along with the corresponding prompt information, are also provided in the supplementary material (Figure A7, Figure A8, and Tables A6–A9).
>
> ## Q7: Detailed LoRA fine-tuning training configuration.
>
> We appreciate the reviewer's feedback. We have provided a description of the training configuration in the main paper (Page 7, Lines 274–280). Below, we further supplement it with additional implementation details.
>
> In the first supervised fine-tuning stage, we set `lora_rank` and `lora_alpha` set to 8 and 32, respectively, applied to all linear layers. The model is trained for 5 epochs on 8 NVIDIA A800 GPUs, with a batch size of 2 per GPU and `gradient_accumulation_steps` set to 8. DeepSpeed ZeRO Stage 2 is employed to optimize memory usage. We set `VIDEO_MAX_PIXELS` to 602112 and `FPS_MAX_FRAMES` to 12 during training.
>
> In the second reinforcement learning stage, we add new LoRA parameters, with `lora_rank` and `lora_alpha` set to 8 and 32. The batch size per GPU is set to 4, with gradient accumulation steps of 8, training for 1 epoch. We utilize vLLM to accelerate inference, generating 8 samples per iteration for GRPO reinforcement learning. Training is conducted using DeepSpeed ZeRO Stage 3.
>
> In the third post-training stage, new LoRA parameters (`lora_rank=8` and `lora_alpha=32`) are added. We randomly select 10,000 samples from RQA-70K for a small-scale supervised fine-tuning. The training parameters are similar to those in the first stage, except that lora_rank is set to 4.
>
> ## Q8: Reasons for low BLEU and METEOR scores.
>
> We appreciate the reviewer's feedback.
>
> First, one reason for the relatively low BLEU scores is the metric's sensitivity to exact n-gram matches. BLEU was originally designed for machine translation. As a result, if the predicted sentences use synonyms or have different word order compared to the references, BLEU may penalize them heavily, even if the semantic meaning is similar[1].
>
> Second, the inherently challenging nature of our task contributes to lower scores across BLEU and METEOR. Each sample of the assessment tasks contains complex, informative meteorological descriptions, which makes it more difficult to achieve high scores, especially compared to less domain-specific tasks. For example, in WeatherQA[2], a domain-specific task focusing on severe weather reasoning, even after fine-tuning, MLLMs still yield relatively low BLEU and METEOR scores, similar to those in our setting. This further highlights the inherent difficulty of domain-specific tasks for current MLLMs.
>
> Finally, despite the relatively low BLEU and METEOR scores, the overall performance of RadarQA shows consistent improvements over baselines, which demonstrates strong potential for weather quality analysis.
>
> [1] Celikyilmaz A, Clark E, Gao J. Evaluation of text generation: A survey[J]. arXiv preprint arXiv:2006.14799, 2020.
>
> [2] Ma C, Hua Z, Anderson-Frey A, et al. Weatherqa: Can multimodal language models reason about severe weather?[J]. arXiv preprint arXiv:2406.11217, 2024.

---

> > ### Comment · Reviewer_AvwT · 2025-08-05
> > **Thanks for the rebuttals**
> >
> > Thanks to the authors for the detailed rebuttals. I'm increasing my score attributing to authors commitment to release the dataset and codes.

---

> > > ### Author Response · Authors · 2025-08-06
> > >
> > > Thanks for recognizing our work! We are grateful that your concerns have been clarified, and we will carefully revise the manuscript accordingly. If you have additional questions or concerns, we would be more than happy to continue the discussion.

---

### Official Review · Reviewer_78LW · 2025-07-03

**Clarity:** 3
**Significance:** 2
**Originality:** 2
**Rating:** 4
**Confidence:** 1

**Summary:**

This work proposes a new benchmark for multi-modal quality analysis in weather radar forecasting, supporting both frame-wise and sequence-wise evaluation tasks. It introduces a hybrid annotation pipeline that combines large language models (LLMs) and human annotators. The authors also train a model to analyze forecasting performance based on this benchmark.

**Questions:**

See weaknesses

**Ethical Concerns:**

["NO or VERY MINOR ethics concerns only"]

**Final Justification:**

The rebuttal satisfactorily addressed my concerns about this paper. However, I do not fully understand the background of the setting. I will therefore maintain my rating of borderline accept.

**Limitations:**

The proposed method appears to be more focused on dataset construction for weather forecasting rather than introducing a fundamentally new modeling approach.

**Quality:**

3

**Strengths And Weaknesses:**

Strengths:

1. The paper provides a thorough analysis of the collected dataset, which appears well-suited for the weather forecasting task.

2. It explores a diverse set of question types, using various input modalities—including text, single images, and image sequences—to support a range of question-answering tasks.

Weaknesses:

1. The technical contributions beyond dataset collection appear limited.

2. Regarding the model selection, it is unclear whether different types or sizes of foundation models were explored. This makes it difficult to assess the effectiveness and advantages of the proposed training scheme

---

> ### Author Rebuttal · Authors · 2025-07-31
>
> We sincerely thank Reviewer 78LW and the area chair for their efforts and appreciation of our comprehensive dataset analysis for the weather forecasting task, the diversity of question types, and input modalities. We will update the manuscript as suggested. Below, we address the reviewer's main concerns point by point.
>
>
> ## Q1：Contributions beyond dataset collection.
>
> In fact, **the lack of a large-scale, high-quality dataset is the main gap in the field of quality evaluation for nowcasting tasks**. Recently, many MLLM-based assessment methods, such as Q-Instruct[1], Co-Instruct[2], and DepictQA[3], have also emphasized the importance of dataset construction for quality evaluation. In the field of meteorology, existing efforts have largely focused on developing new numerical metrics, while neglecting a quality understanding and perception grounded in expert evaluation practices. In contrast, our proposed dataset, RQA-70K, addresses this gap by incorporating both frame-level and sequence-level data modalities and supporting both rating-based and assessment-based evaluation tasks, thereby bridging numerical evaluation metrics and expert-driven analysis approaches.
>
> Besides the collected dataset, our contributions mainly consist of three aspects:
> 1. **A novel quality evaluation paradigm for meteorology.** By integrating the evaluation steps of meteorological experts and existing metric-based quality assessment methods, we propose four quality analysis tasks: *frame rating, frame assessment, sequence rating, and sequence assessment*. These tasks cover both frame-level and sequence-level modalities, as well as rating-based and assessment-based evaluation needs.
> 2. **A well-designed data construction pipeline**. To construct our dataset, we begin by selecting seven classical nowcasting models based on various backbones to generate the RawRQA-20K dataset by using storm events from SEVIR. Second, in collaboration with domain experts, we establish a *scientific attribute library* containing a total of 37 attributes. Based on the characteristics of these attributes, we adopt a hybrid labeling strategy that combines automated and manual annotation on RawRQA-20K. Finally, we carefully design instruction prompts to guide GPT-4o in generating data for *assessment tasks* based on these attributes, and we use scripts to construct JSON-formatted data for *rating tasks*, ultimately resulting in the RQA-70K dataset.
> 3. **A multi-stage training pipeline.** To train a high-performance quality evaluation model, we first employ supervised fine-tuning (SFT) to equip the model with foundational capabilities. Then, we adopt GRPO along with two carefully designed reward functions on the two rating tasks. Finally, we conduct an additional round of SFT on a small subset of RQA-70K with a lower LoRA rank to further enhance the model's performance.
>
> [1] Wu H, et al. Q-Instruct: Improving low-level visual abilities for multi-modality foundation models, CVPR 2024.
>
> [2] Wu H, et al. Towards open-ended visual quality comparison, ECCV 2024.
>
> [3] You Z, et al. Depicting beyond scores: Advancing image quality assessment through multi-modal language models, ECCV 2024.
>
> ## Q2：More foundation models.
>
> We adopt Qwen2.5-VL-7B-Instruct as our base model, following many recent quality evaluation studies [1,2,3]. To provide a more comprehensive comparison, here we additionally conduct experiments using Qwen2.5-VL-3B-Instruct. The experimental results below show that the 3B model shows a slight drop in performance while using fewer parameters.
>
> | Accuracy                    | Overall | False Alarm | Miss   | High Value | Sharpness | Overall | Dynamic Consistency | Cumulate Precipitation | High Value Retain |
> |----------------------------|---------|-------------|--------|-------------|-----------|---------|----------------------|-------------------------|--------------------|
> |          | Frame|Frame|Frame|Frame|Frame|Sequence|Sequence|Sequence|Sequence|
> | Qwen2.5-VL-3B Instruct     | 60.93   | 58.95       | 57.33  | 65.81       | 74.19     | 64.42   | 53.43                | 46.19                   | 74.78              |
> | Qwen2.5-VL-7B Instruct     | 61.51   | 65.35       | 67.67  | 69.19       | 78.60     | 66.17   | 53.31                | 48.94                   | 80.52              |
>
>
>
>
> | Frame / Sequence               | BERTScore     | BLEU          | METEOR        | ROUGE_L       | GPT4Score   |
> | ---------------------- | ------------- | ------------- | ------------- | ------------- | ----------- |
> | Qwen2.5-VL-3B Instruct |0.811 / 0.815       |0.212 / 0.216       |0.425 / 0.465       |0.513 / 0.434       |6.77 / 6.36      |
> | Qwen2.5-VL-7B Instruct | 0.809 / 0.815 | 0.213 / 0.212 | 0.420 / 0.461 | 0.512 / 0.436 | 6.87 / 6.58 |
>
>
> [1] Li W, Zhang X, Zhao S, et al. Q-Insight: Understanding image quality via visual reinforcement learning[J]. arXiv preprint arXiv:2503.22679, 2025.
>
> [2] Wu T, Zou J, Liang J, et al. VisualQuality-R1: Reasoning-Induced Image Quality Assessment via Reinforcement Learning to Rank[J]. arXiv preprint arXiv:2505.14460, 2025.
>
> [3] Zhang X, Li W, Zhao S, et al. VQ-Insight: Teaching VLMs for AI-Generated Video Quality Understanding via Progressive Visual Reinforcement Learning[J]. arXiv preprint arXiv:2506.18564, 2025.

---

> ### Author Response · Authors · 2025-08-06
>
> Dear reviewer 78LW:
>
> Thank you again for your valuable comments on our submission. As the discussion phase is approaching its end, we would like to kindly confirm whether we have addressed all of your concerns. Should there be any remaining questions requiring further clarification, please do not hesitate to let us know.
>
> We sincerely look forward to your feedback.

---

### Comment · Area_Chair_rgno · 2025-08-03

Dear Reviewers,

Please take a moment to revisit the paper, evaluate the authors’ reponses, and confirm whether the updates address any concerns raised during your initial review. If you have already completed this step, thank you for your swift action. Otherwise, we appreciate you doing so as soon as possible.
Thank you again for your valuable time and contributions!

Thanks，

AC

---

### Note · Authors · 2025-08-15

Dear all reviewers and area chair:

We sincerely thank the AC and all reviewers for providing us the opportunity to further present our work, for the reviewers' thoughtful engagement, and for the AC's guidance throughout the process. We are encouraged that three reviewers have indicated that their concerns have been addressed, while one reviewer has not provided additional feedback after the initial round.
* For Reviewer AvwT and Reviewer 6kbe, they expressed appreciation from the beginning, and after further discussions on topics such as comparisons with stronger domain-specific baselines and the public availability of the proposed dataset and code, they offered greater appreciation for our work.
* Reviewer Htbn engaged in active exchange with us and finally aligned with our perspective, noting "The rebuttal addressed my concern." and "I have updated my score".
* For reviewer 78LW, we provided detailed responses addressing the points on contributions beyond dataset collection and the inclusion of more foundation models, and we believe these have adequately resolved the reviewer's concerns.

During rebuttal, we also committed to:
1. releasing our code and dataset.
2. conducting a larger-scale expert study to make our results more convincing.
3. elaborating in the limitation section on the bias introduced by using GPT-4o for organizing annotated attributes.
4. improving the paper's presentations by refining formatting and figure captions.

Finally, we thank the area chair and all reviewers again for their recognition and efforts devoted to the process.

Best regards,

Authors of submission #2705

---

### Decision · Program_Chairs · 2025-09-17

**Decision:**

Accept (poster)

**Comment:**

Summarize the scientific claims and findings:

This paper proposes RadarQA, a multi-modal large language model (MLLM)-based framework for quality analysis of weather radar forecasts. The authors define a four-task evaluation paradigm (frame-level vs. sequence-level × rating vs. assessment) that integrates quantitative scoring with descriptive assessment, aiming to bridge the gap between traditional numerical metrics (e.g., CSI, POD, FAR) and expert-driven meteorological evaluations. They introduce RQA-70K, a large-scale dataset created via a hybrid annotation pipeline that combines human expert labeling, automated heuristics, and GPT-4o-assisted text generation. A three-stage training strategy (supervised fine-tuning, GRPO-based reinforcement learning, and post-training) is applied to a Qwen2.5-VL backbone, resulting in performance gains over both general-purpose MLLMs (e.g., GPT-4o, Claude-3.7) and domain-specific baselines, including in out-of-distribution settings such as radar super-resolution tasks.

Strengths of the paper:

- Addresses a real gap between numerical metrics and expert meteorological interpretation, with well-articulated rationale and domain-specific grounding.

- The four-task matrix closely mirrors expert workflows, covering both static and dynamic aspects of radar forecast quality.

- The construction of the RQA-70K dataset is a substantial contribution. The hybrid annotation pipeline, combining expert-defined physical attributes, automated scripts, and LLM-assisted text generation, is a robust and scalable approach for creating high-quality, domain-specific data.

- The paper includes extensive experiments, including ablations of the training pipeline, comparisons against strong open- and closed-source MLLM baselines, and out-of-distribution tests. The results are consistently positive and convincing.

Weaknesses:

- Clarity of rl training details: The design and justification of the specific reward functions for the RL stage, while addressed in the rebuttal, could be more thoroughly explained and analyzed in the main paper to alleviate concerns about their simplicity.

- Algorithmic novelty: The core technical approach adapts existing MLLM fine-tuning and reinforcement learning techniques (SFT, GRPO) to a new domain.

Reasons for decision:

The decision to Accept this submission as a poster is based on its significant contributions to applied AI and the meteorological sciences. The primary reasons are: (1) The work directly enables a valuable application (expert-level forecast analysis). It has clear potential for real-world impact in weather prediction and safety. (2) The creation and promised release of the RQA-70K dataset is a major benefit to the research community that will facilitate future work in this area. (3) The model demonstrates clear and substantial improvements over very strong baselines across a comprehensive set of tasks, validated by both automated metrics and human experts.

Summary of rebuttal and discussion:

The author rebuttal was comprehensive and effectively addressed the majority of reviewer concerns: (1) The authors clarified that their contributions include a novel evaluation paradigm and a multi-stage training pipeline, contextualizing their dataset effort alongside recent trends in quality-focused MLLM research. (2) Reviewers requested more details on model selection, LoRA configuration, and RL rewards. The authors provided extensive additional experimental results, full training hyperparameters, and a detailed mathematical breakdown of the reward functions, satisfying these queries. (3)  A key concern was the lack of public code/data and the use of GPT-4o for annotations. The authors made a firm commitment to release all code, data, and model weights upon acceptance, which was a decisive factor for one reviewer to increase their score. They also acknowledged potential biases from GPT-4o and agreed to discuss them in the limitations. (4) A request for comparisons against traditional meteorological metrics was fully addressed with new results showing RadarQA's superior alignment with expert judgment.

The rebuttal process strengthened the submission. Initial concerns about scope, reproducibility, and evaluation were convincingly mitigated. The reviewers converged towards a positive consensus, recognizing the work's value as a solid technical contribution with high practical utility and a valuable resource for the community.